# Fasting mimicking diet in mice delays cancer growth and reduces immunotherapy-associated cardiovascular and systemic side effects

S. Cortellino [1,2,10], V. Quagliariello[3,10], G. Delfanti[4], O. Blaževitš[1], C. Chiodoni [5], N. Maurea[3], A. Di Mauro[6], F. Tatangelo [6], F. Pisati[7], A. Shmahala[1], S. Lazzeri[1], V. Spagnolo[1], E. Visco[1], C. Tripodo [1,8], G. Casorati [4], P. Dellabona [4] & V. D. Longo [1,9] ✉

Immune checkpoint inhibitors cause side effects ranging from autoimmune endocrine disorders to severe cardiotoxicity. Periodic Fasting mimicking diet (FMD) cycles are emerging as promising enhancers of a wide range of cancer therapies including immunotherapy. Here, either FMD cycles alone or in combination with anti-OX40/anti-PD-L1 are much more effective than immune checkpoint inhibitors alone in delaying melanoma growth in mice. FMD cycles in combination with anti-OX40/anti-PD-L1 also show a trend for increased effects against a lung cancer model. As importantly, the cardiac fibrosis, necrosis and hypertrophy caused by immune checkpoint inhibitors are prevented/reversed by FMD treatment in both cancer models whereas immune infiltration of CD3[+] and CD8[+] cells in myocardial tissues and systemic and myocardial markers of oxidative stress and inflammation are reduced. These results indicate that FMD cycles in combination with immunotherapy can delay cancer growth while reducing side effects including cardiotoxicity.

Immune checkpoint inhibitors (ICIs) boost anti-tumor immune response by mitigating the self-tolerance mechanism of the immune cells, which is hijacked by tumor cells. The clinical application of ICI has profoundly improved prognosis and life expectancy in metastatic cancer patients suffering from melanoma, non-small-cell lung, and kidney cancer, representing a paradigm shift in cancer therapy[1–3]. Immunotherapy is based on the role of cell-surface receptors and ligands accessory to the T-cell receptor in inhibiting cell-mediated immune response. The first monoclonal antibodies targeting this inhibitory axis were developed against the immune checkpoint PD-1 (programmed death-1), its ligand PD-L1 (programmed death ligand-1) and CTLA-4 (cytotoxic T lymphocyte antigen-4). They were first tested in the treatment of melanoma and then applied to the treatment of other cancers characterized by poor prognosis[4–12]. Although ICIs therapy has improved the survival of many cancer patients, the percentage of patients responding remains low. In order to improve

[1]IFOM, The AIRC Institute of Molecular Oncology, 20139 Milan, Italy. [2]Laboratory of Pre-Clinical and Translational Research, IRCCS-CROB, Referral Cancer Center of Basilicata, 85028 Rionero in Vulture, Italy. [3]Division of Cardiology, Istituto Nazionale Tumori-IRCCS-Fondazione G. Pascale, Naples, Italy. [4]Experimental Immunology Unit, Division of Immunology, Transplantation and Infectious Diseases, IRCCS San Raffaele Scientific Institute, Milan, Italy. [5]Department of Experimental Oncology, Fondazione IRCCS Istituto Nazionale Tumori, Milan, Italy. [6]Pathology and Cytopathology Unit, Department of Support to Cancer Pathways Diagnostics Area, Istituto Nazionale Tumori-IRCCS "Fondazione G. Pascale", 80131 Naples, Italy. [7]Histopathology Unit, Cogentech Società Benefit srl, 20139 Milan, Italy. [8]University of Palermo School of Medicine, Palermo, Italy. [9]Longevity Institute and Davis School of Gerontology, University of Southern California, Los Angeles, CA 90089, USA. [10]These authors contributed equally: S. Cortellino, V. Quagliariello. ✉e-mail: vlongo@usc.edu

efficacy and patient response rates, new therapeutic strategies combining ICIs with adjuvants that augment immune-dependent attack of cancer cells are needed. For example, targeting alternative pathways such as the co-stimulatory molecules OX40, 4-1BB, glucocorticoid-induced TNFR-related protein (GITR) has proven to enhance T-cell mediated immunity in preclinical model[13–17], although no clinical studies have confirmed the efficacy of such treatments in humans.

However, ICIs also causes side effects which are uniquely associated with an increase in autoimmunity due to alteration of self-tolerance. A recent study found that 3.5% of patients initiating ICI experience adverse events requiring hospitalization and immunosuppression[18,19]. Immune-related adverse events (IRAEs) can affect colon, lungs, liver, skin, pituitary, thyroid, and heart[20]. Although cardiotoxicity accounts for <1% of IRAEs, the onset of such complications, such as myocarditis, arrhythmia, pericarditis and vasculitis results in death in 50% of cases[21].

Cardiovascular immune-related adverse events include myocarditis, pericardial disease, vasculitis, Takotsubo syndrome, destabilization of atherosclerotic lesions, venous thromboembolism, and conduction abnormalities[22]. ICI-associated myocarditis results from inflammation of the conduction system due to infiltration of T cells and macrophages[23]. Under physiological conditions, the cardiac lymphocyte infiltrate is limited and the macrophages and dendritic cells resident in the heart control the expression of the immune checkpoint proteins in order to maintain homeostasis. Thus, inhibiting the immune checkpoint pathway could lead to adverse outcomes, by promoting the recruitment of lymphocytes and macrophages and triggering an inflammatory response[24].

Indeed, in genetically modified mice, deletion of CTLA4 leads to massive lymphoproliferative disease and diffuse lymphocyte infiltration in almost all organs, including the heart. In contrast the deletion of Pdcd1 (encoding PD-1) in Balb/c mice causes cardiomyopathy due to the development of autoantibodies against troponin I[25]. On the other hand, activation of OX40 with agonistic antibodies stimulates the release of proinflammatory cytokines (IL6, TNFα, IFNγ) by activated T lymphocytes and antigen-presenting cells (APCs)[26,27] thus causing a systemic inflammatory response syndrome[26,28,29]. Thus, it is important to develop new strategies capable of increasing the anticancer efficacy of immunotherapy while preventing unwanted side effects.

In recent years, periodic cycles of fasting or fasting mimicking diets (FMDs) have emerged as effective in potentiating the anti-cancer effects of chemotherapy, hormone therapy, and kinase inhibitors against cancer cells while reducing side effects in mice[30–35]. More recent work indicates that fasting can also potentiate immunotherapy against lung cancer[36] and breast cancer[37] in agreement with the role of fasting/FMD in combination with chemotherapy in increasing the T cell-dependent attack of breast cancer and melanoma cells[38,39].

In fact, prolonged fasting and a low calorie FMDs have been shown to have a major impact on the regulation and renewal of the immune system[40–43]. The metabolic and physiologic changes that result from prolonged fasting promote hematopoietic stem cell (HSC) enrichment in the bone marrow and lead to lymphoid and myeloid population migration from the peripheral blood to the bone marrow[41,44,45]. In this favorable environment, fasting rejuvenates HSC, improves memory T cell function and strengthens the immune responses by stimulating autophagy or apoptosis, which can remove damaged organelles, molecules and cells[44]. At the same time fasting can reduce monocytes proinflammatory activity by shifting their metabolism from glycolysis to oxidative phosphorylation (OXPHOS)[45]. These effects of fasting on the immune system are mediated, in part, by the modulation of the IGF-1-PKA nutrient sensing pathway[41].

Alternating fasting/FMD cycles with normal nutrition can also reverse or ameliorate autoimmune diseases including multiple sclerosis and inflammatory bowel disease in mouse models by reducing inflammation, removing autoimmune cells, and increasing

hematopoietic stem cells, which generate differentiated immune cells from progenitor/stem cells[46,47].

Notably, fasting/FMD cycles promotes immune cell infiltration and delays tumor growth in both breast cancer and melanoma cells, raising the possibility that it could enhance the efficacy of immunotherapy without increasing side effects[38]. In this study we tested whether FMD in combination with anti-OX40/anti-PD-L1 or anti-PD-1/antiCTLA-4 improves the antitumor immune response against B16-F10 melanoma and LLC1 lung tumor and investigated its effect on cardiac adverse events. We show that FMD can delay tumor growth of B16-F10 melanoma but only causes non significant trends for improved anti-cancer effects of anti-OX40/anti-PD-L1. FMD cycles, also prevents the cardiotoxicity of ICIs in both the melanoma and lung cancer models.

## Results
### One cycle of FMD is not sufficient to potentiate the effects of PD-1 and CTLA4 against B16F10 melanoma tumors
To determine whether FMD potentiates the efficacy of immunotherapy in low immunogenic tumors such as melanoma, we tested FMD cycles lasting 4 days in combination with immunotherapy directed against the immune checkpoints PD-1 and CTLA4, successfully used in the clinic for the treatment of patients with melanoma. B16F10 cells were injected subcutaneously into C57BL/6 mice and after three days mice were subjected to one FMD cycle and treated with anti-PD-1 and anti-CTLA4 on day 4, 6 and 8 according to the scheme shown in Fig. 1A. The combined treatment was performed simultaneously as used in the clinic. Combined anti-PD-1/anti-CTLA4 therapy only caused a trend for delayed growth of B16F10 tumors compared to the untreated groups (Fig. 1B, C). One cycle of FMD also was not sufficient to slow tumor growth in either the control or immunotherapy groups compared with the ad libitum (AL) diet. However, the analysis of the immune infiltrate 10 days after refeeding, shows that ICIs therapy alone increases the levels of cytotoxic $CD8^+CD44^+GzmB^+$ lymphocytes (Fig. 1D), effector memory $CD8^+CD44^+CD62L^-$ T cells (Fig. 1E), and cytotoxic $CD45^+Nkp46^+GzmB^+$ NK cells (Fig. 1F) independently of the diet.

FMD cycles combined with anti-PD-1/anti-CTLA4 caused a trend for an increase (statistically not significant) of both cytotoxic T cells and NK cells into tumor tissue compared to the standard diet group treated with ICIs (Fig. 1D, F). IHC analyses of tumor sections confirmed that immunotherapy increases the infiltration of CD8 T cells (Supplementary Fig. 2A), whereas FMD has no effect on CD8 T cell tumor infiltration. CD4 T cells, B cells and myeloid cells were not affected by diet and immunotherapy (Supplementary Fig. 2B–D), as observed by FACS analysis.

Since previous studies showed that three FMD cycles in combination with chemotherapy can control breast cancer and melanoma growth by an immunity-dependent mechanism[37,38], we tested whether additional cycles of FMD are required to observe effects on tumor growth and immune cells.

### Two cycles of FMD delay B16F10 melanoma growth and show a trend for improving the efficacy of anti-OX40/anti-PD-L1 therapy
Next, we have tested whether two cycles of FMD enhances the efficacy of another immunotherapy treatment in melanoma by combining antagonist antibody against immune checkpoint PD-L1 and agonist antibody against costimulatory molecule OX40.

Mice were injected subcutaneously with B16F10 melanoma cells and subjected to two cycles of FMD combined with three doses of anti OX40 and anti PD-L1. To avoid reported counterproductive effects, such as cytokine storm and T cell apoptosis[48], ICIs therapy was administered sequentially: on days 4, 6 and 8 anti-OX40 and on day 11, 13 and 15 anti-PD-L1, (Fig. 2A).

Whereas anti-OX40/anti-PD-L1 therapy had no effect on melanoma growth, two cycles of FMD plus anti-OX40/anti-PD-L1 caused a strong delay of tumor progression, although most of the effect

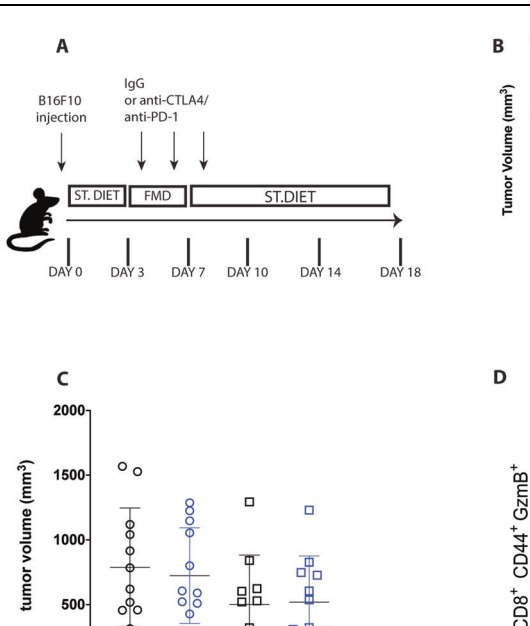

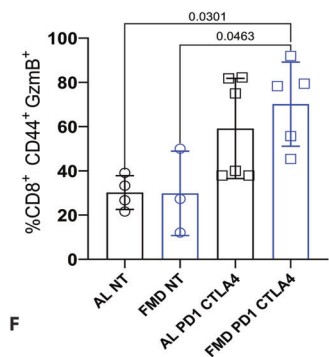

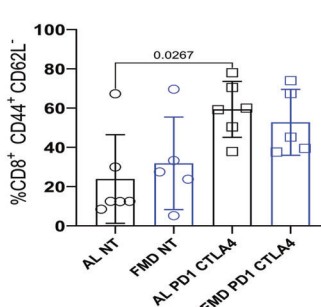

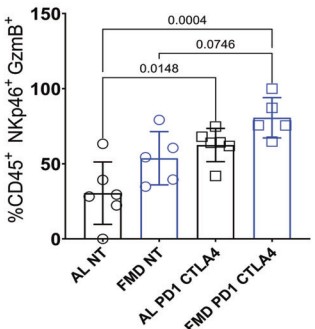

**Fig. 1 | FMD does not improve the efficacy of anti-PD-1/anti-CTLA4 combo therapy against B16 F10 melanoma tumor. A** Schedule of tumor implantation and treatment for B16F10 syngeneic tumor models. **B**, **C** B16 Tumor growth in immunocompetent C57/BL6 syngeneic mice treated with isotype control and anti-PD-1/anti-CTLA4 and fed with standard diet or FMD (AL NT $n = 12$; FMD NT $n = 12$; AL PD1 CTLA4 $n = 10$; FMD PD1 CTLA4 $n = 11$). Analysis of tumor immune infiltrate by FACS: **D** CD8$^+$ CD44$^+$GzmB$^+$ cytotoxic T cell (AL NT $n = 4$; FMD NT $n = 3$; AL PD1 CTLA4 $n = 6$; FMD PD1 CTLA4 $n = 5$); **E** CD8$^+$ CD44$^+$CD62L$^-$ effector memory T cell (AL NT $n = 6$; FMD NT $n = 5$; AL PD1 CTLA4 $n = 6$; FMD PD1 CTLA4 $n = 5$); **F** CD45$^+$NKp46$^+$GzmB$^+$ NK cell (AL NT $n = 6$; FMD NT $n = 5$; AL PD1 CTLA4 $n = 6$; FMD PD1 CTLA4 $n = 5$). Statistical analysis was performed using one-way analysis of variance (ANOVA). P values were determined by one-way ANOVA with Tukey's post analysis Differences were considered significant when $P \le 0.05$. All data are represented as mean ± SEM. Source data are provided as a Source Data file.

appears to be caused by the dietary intervention since FMD plus anti-OX40/anti-PD-L1 only caused a non significant trend for improved anti-cancer effects compared to FMD alone (Fig. 2B, C).

To determine if FMD and immunotherapy affect the tumor microenvironment, we analyzed tumor infiltrating lymphocytes (TILs) by FACS and IHC after the refeeding (Fig. 2, Supplementary Fig. 2). Surprisingly in B16F10 melanoma tumors, anti-OX40/anti-PD-L1 treatment alone, independently of the diet, increased the percentage of total T lymphocytes (CD45$^+$CD3$^+$) (Fig. 2D, Supplementary Fig. 2A) and T helper cells (CD45$^+$CD3$^+$CD4$^+$) (Supplementary Fig. 3A), and promoted the activation of cytotoxic effector T cell (CD3$^+$CD8$^+$CD44$^+$GzmB$^+$) (Fig. 2E) while reducing the T reg cell population (CD3$^+$CD4$^+$CD25$^+$) (Supplementary Fig. 3B). IHC staining of tumor section with CD4 and B220 antibody revealed no differences in the distribution of CD4 and B cells within the tumor bed (Supplementary Fig. 2B–C).

Mice treated with the combination of ICIs and FMD showed only a trend for increased accumulation of cytotoxic NK cells (CD45$^+$Nkp46$^+$GzmB$^+$) inside the tumor (Fig. 2F) and increased expression of CD127 by CD8$^+$CD44$^+$ effector T cell compared to ICI alone (Supplementary Fig. 3C). These potentially higher levels of CD127 on effector T cell could indicate better homeostasis and greater long-term anti-tumor memory protective function.

Regarding the innate immune system, FMD alone increases the percentage of dendritic cells (CD11c$^+$ MHCII$^+$) (Fig. 3A) and macrophages (CD45$^+$ CD11b$^+$ F4/80$^{high}$) (Fig. 3B), an effect reversed by ICI drugs.

Both FMD, anti-OX40/anti-PD-L1 treatment and their combinations reduce the percentage of the immunosuppressive M-MDSC (CD11b$^+$Ly6C$^{high}$) in the tumor bed compared to the standard diet group (Fig. 3C), whereas PMN-MDSC (CD11b$^+$Ly6C$^{low}$Ly6G$^{high}$)

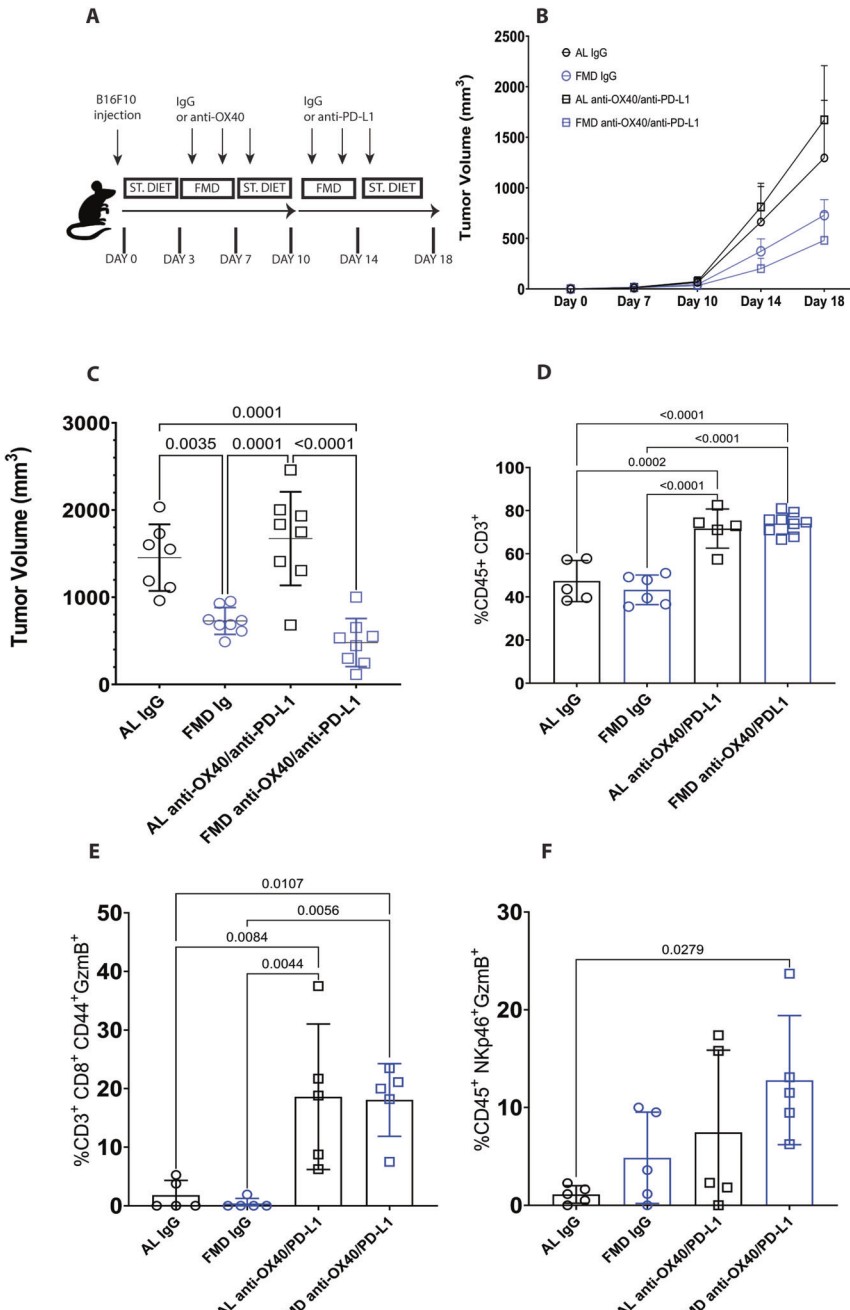

**Fig. 2 | FMD delays B16 tumor growth and activates NK cells. A** Schedule of tumor implantation and treatment for B16F10 syngeneic tumor models. **B**, **C** B16 Tumor growth in immunocompetent C57/j syngeneic mice treated with isotype control and anti-OX40/anti–PD-L1 and fed with standard diet or FMD (AL NT $n = 7$; FMD NT $n = 8$; AL anti-OX40/anti-PD-L1 $n = 8$; FMD anti-OX40/anti-PD-L1 $n = 8$). **D** CD45+CD3+ T cell (AL NT $n = 5$; FMD NT $n = 6$; AL anti-OX40/anti-PD-L1 $n = 5$; FMD anti-OX40/anti-PD-L1 $n = 9$), **E** CD3+CD8+GzmB+ cytotoxic effector memory T cells

(AL NT $n = 5$; FMD NT $n = 5$; AL anti-OX40/anti-PD-L1 $n = 5$; FMD anti-OX40/anti-PD-L1 $n = 5$, **F** CD45+NKp46+GzmB+ NK cells (AL NT $n = 5$; FMD NT $n = 5$; AL anti-OX40/anti-PD-L1 $n = 5$; FMD anti-OX40/anti-PD-L1 $n = 5$). Statistical analysis was performed using one-way analysis of variance (ANOVA). *P* values were determined by one-way ANOVA with Tukey's post analysis. Differences were considered significant when $P \leq 0.05$. All data are represented as mean ± SEM. Source data are provided as a Source Data file.

display a trend for enrichment only in the AL group treated with ICIs (Fig. 3D).

These results suggest that FMD may favor the priming of T cells by recruiting dendritic cells into the tumor bed, reducing the percentage of immunosuppressive M-MDSC and promoting the cytolytic activity of NK cells.

However, the reduction of the tumor mass observed in mice subjected to FMD might also be explained by a direct effect of FMD on the tumor cells through the regulation of oncogenic pathways or by modification of tumor metabolism[35,37].

## FMD reduces cardiac fibrosis, necrosis and hypertrophy in melanoma-bearing mice treated with immune-checkpoint inhibitors

Cardiotoxicity due to ICIs is uncommon, with an incidence of approximately 1%[21] but it is in most cases severe and can be life-threatening. Patients can present with cardiac fibrosis, cardiac arrest, autoimmune myocarditis, cardiomyopathy, heart failure, pericardial involvement, and vasculitis[22]. Endomyocardial biopsy is the gold standard for diagnosis of ICI-induced myocarditis; it typically shows lymphocyte and macrophage infiltration with myocardial fibrosis.

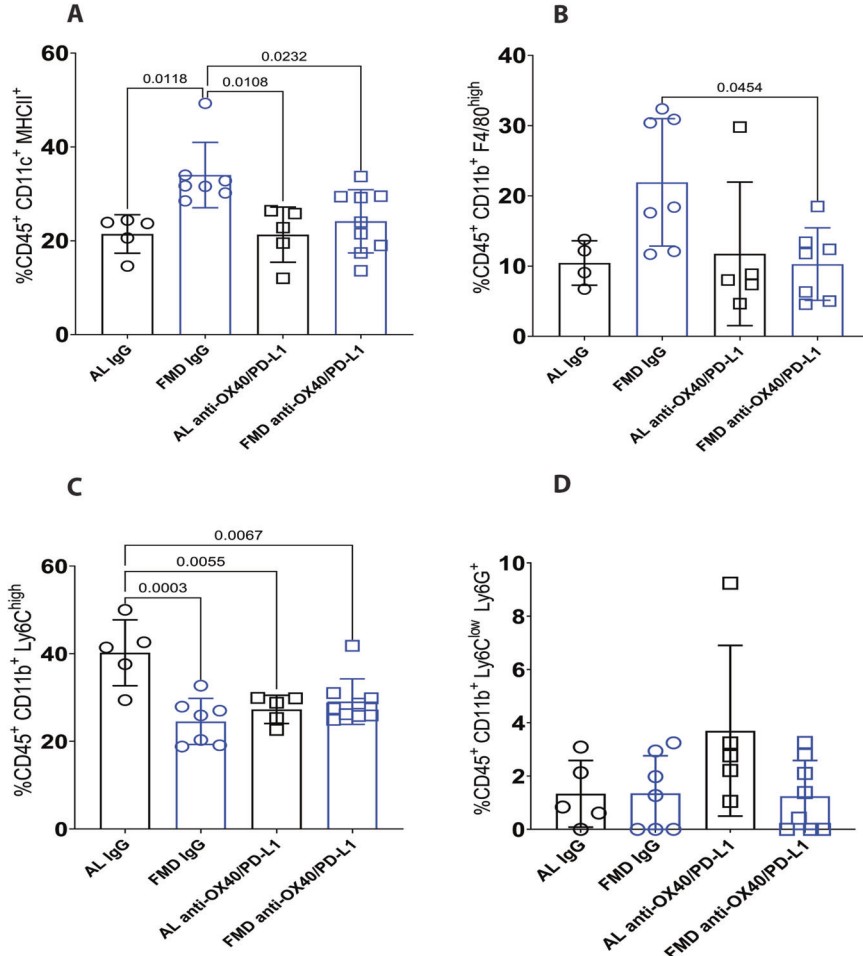

**Fig. 3 | FMD modulates tumor infiltrating myeloid cells. A** CD45+CD11c+MHCII+ dendritic cell (AL NT $n = 5$; FMD NT $n = 7$; AL anti-OX40/anti-PD-L1 $n = 5$; FMD anti-OX40/anti-PD-L1 $n = 9$); **B** CD45+CD11b+F4/80high macrophage (AL NT $n = 4$; FMD NT $n = 7$; AL anti-OX40/anti-PD-L1 $n = 5$; FMD anti-OX40/anti-PD-L1 $n = 7$); **C** CD45+CD11b+Ly6Chigh M-MDSC (AL NT $n = 5$; FMD NT $n = 7$; AL anti-OX40/anti-PD-L1 $n = 5$; FMD anti-OX40/anti-PD-L1 $n = 9$); **D** CD45+CD11b+Ly6ClowLy6Ghigh PMN-MDSC (AL NT $n = 5$; FMD NT $n = 7$; AL anti-OX40/anti-PD-L1 $n = 5$; FMD anti-OX40/anti-PD-L1 $n = 8$). Statistical analysis was performed using one-way analysis of variance (ANOVA). $P$ values were determined by one-way ANOVA with Tukey's post analysis. Differences were considered significant when $P \leq 0.05$. All data are represented as mean ± SEM. Source data are provided as a Source Data file.

Therefore, we tested whether FMD has beneficial effect on cardiac fibrosis in mice treated with combinatorial ICIs therapies. Fibrosis and necrosis, measured by H&E staining on heart sections, increased in the standard diet + immunotherapy group; an effect reduced or reversed by FMD cycles (one cycle of FMD for anti-PD1-antiCTLA4 and two cycles of FMD for anti-OX40-anti-PD-L1, as shown in scheme of Fig. 1A, B) (Fig. 4A, B).

Then we measured the deposition of procollagen 1α1, another common marker of cardiac fibrosis. The combined immunotherapy (anti-OX40/anti-PD-L1 and anti-PD-1/anti-CTLA-4) in the standard diet group (AL) caused a major increase in the deposition of procollagen 1α1; an effect reduced or reversed by FMD (Fig. 4C). (FMD anti-OX40/anti-PD-L1 5.74 ± 2.42 vs to AL anti-X40/anti-PD-L1 13.74 ± 2.12 ng/mg of total protein; FMD anti-PD-1/anti-CTLA-4 8.34 ± 1.89 vs to AL anti-PD-1/anti-CTLA-4 16.53 ± 1.66; $p < 0.001$) (Fig. 4C).

Metalloproteases type-9 (MMP-9) expression, another biomarker of cardiac fibrosis and heart failure, was also reduced by FMD treatment in mice also treated with anti-OX40/anti-PD-L1 and anti-PD-1/anti-CTLA-4 group (FMD anti-OX40/anti-PD-L1 417.36 ± 102.5 vs AL anti-X40/anti-PD-L1 737.39 ± 102.5 pg/mg of total protein; FMD anti-PD-1/anti-CTLA-4 521.1 ± 88,4 vs to AL anti-PD-1/anti-CTLA-4 816.2 ± 66.7; $p < 0.001$) (Fig. 4C).

These results indicate that FMD reduces collagen type II accumulation in cardiac tissues, contributing to reducing the cardiac dysfunction associated with ICIs. In addition, cardiac quantification of heart mouse pro-collagen 1α1 (ng/mg of total protein) and MMP-9 indicates that FMD can reduce cardiac fibrosis compared to the standard diet group.

## FMD reduces CD3 and CD8 immune infiltration in heart tissues of melanoma-bearing mice treated with combination ICIs therapies

T-cells infiltration in cardiac tissue is the leading cause of myocarditis and pericarditis seen in cancer patients treated with ICIs in monotherapy or in combination with other immunotherapies. The CD3+ and CD8+ T-cell infiltration in cardiac tissue increases inflammation and reduces cardiac functions by activating the MyD88-NLRP3 signaling pathway which plays an important role in the development of myocardial inflammation. In line with the literature, immunotherapy causes a major increase in the cardiac infiltration of CD3+ and CD8+ T-cells in mice fed with a standard diet (Fig. 5A, B). Immune CD3+ and CD8+ staining confirms that ICIs increased lymphocyte cardiac accumulation (Fig. 5A, B). Notably, FMD reduces the expression of CD3+ and CD8+ T cells in mice treated with anti-OX40/anti-PD-L1 and anti-PD-1/anti-CTLA-4 (one cycle

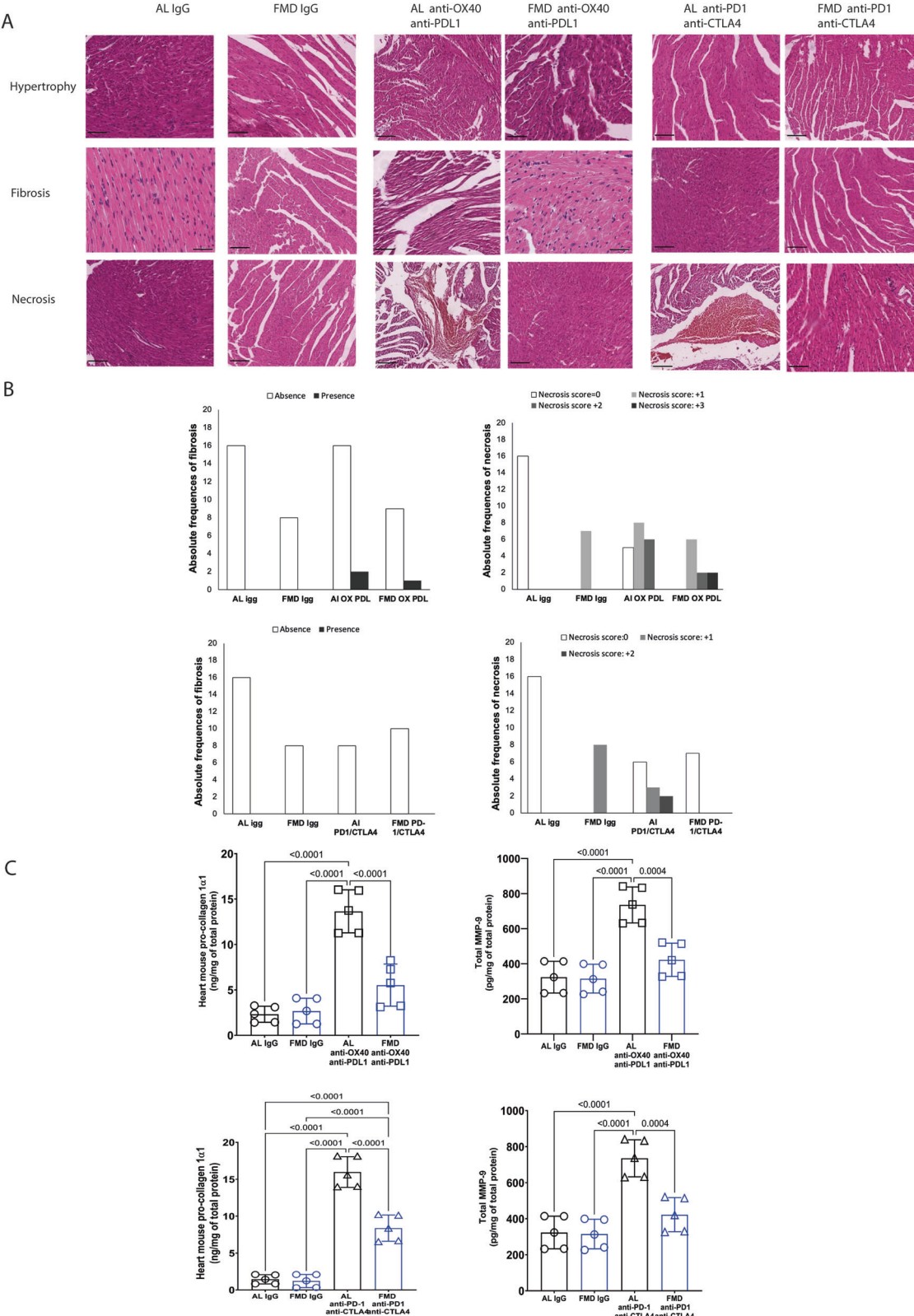

**Fig. 4 | FMD alleviates cardiac fibrosis, necrosis and hypertrophy in C57/j mice bearing B16-F10 melanoma tumor. A** H&E staining in the heart of C57/J mice bearing B16-F10 melanoma tumor and treated with IgG or anti-OX40/anti-PD-L1 or anti-PD-1/anti-CTLA-4. **B** Quantification of heart fibrosis and necrosis in the different experimental group (3 tissue sections for each tumor; 5 tumors per each group). **C** Quantification of pro collagen 1α1 and MMP9 in the heart belonging to different experimental group ($n = 5$). Statistical analysis was performed using one-way analysis of variance (ANOVA). Differences were considered significant when $P < 0.05$. All data are represented as mean ± SEM. Source data are provided as a Source Data file.

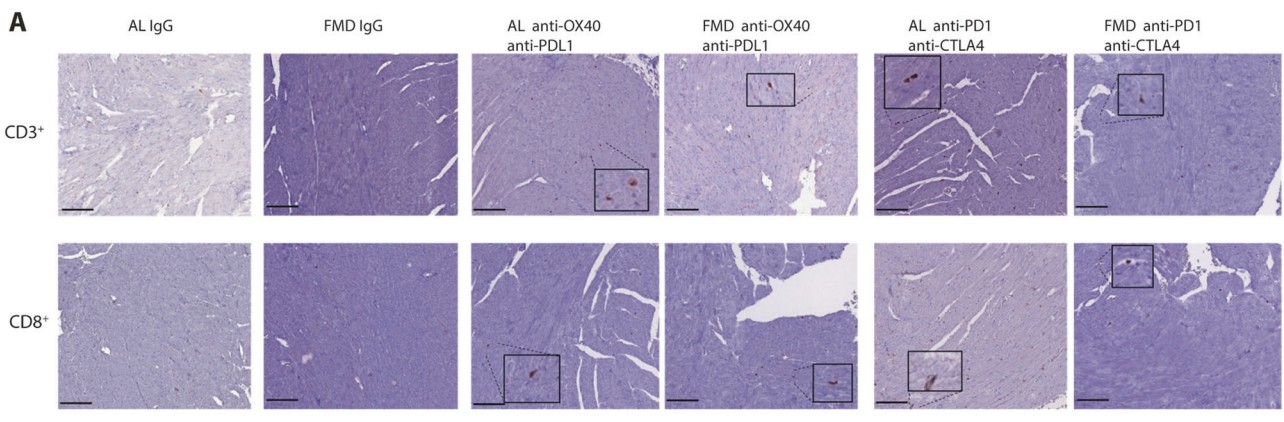

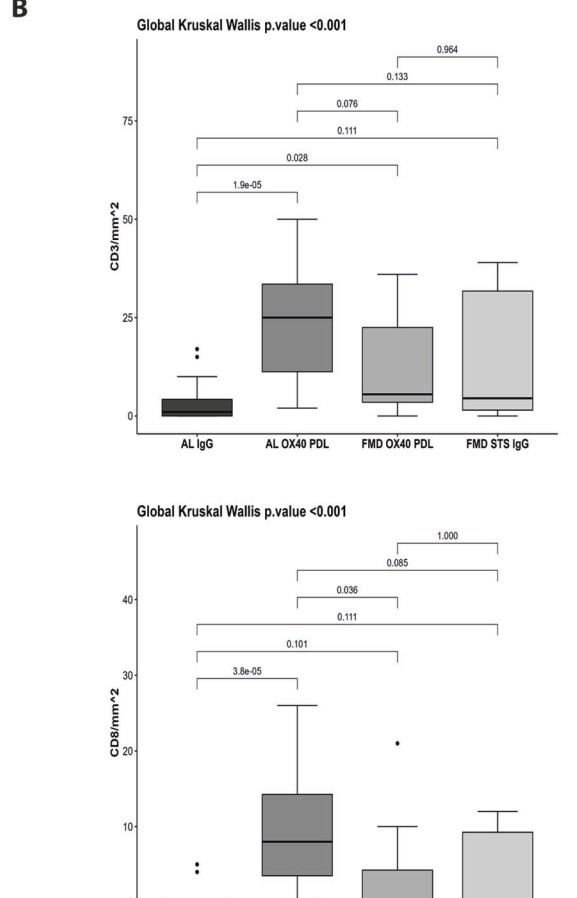

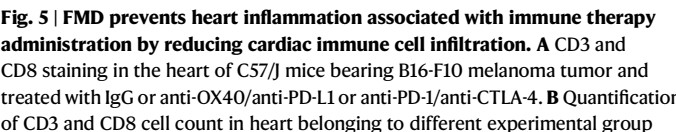

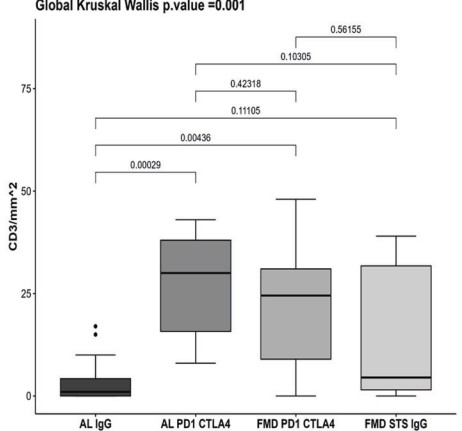

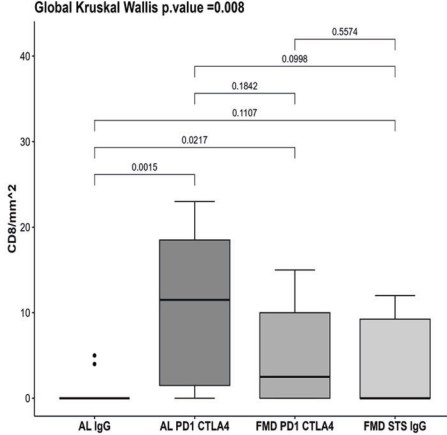

**Fig. 5 | FMD prevents heart inflammation associated with immune therapy administration by reducing cardiac immune cell infiltration. A** CD3 and CD8 staining in the heart of C57/J mice bearing B16-F10 melanoma tumor and treated with IgG or anti-OX40/anti-PD-L1 or anti-PD-1/anti-CTLA-4. **B** Quantification of CD3 and CD8 cell count in heart belonging to different experimental group (3 tissue sections for each tumor; 5 tumors for each group). Global Kruskal Wallis test with post hoc Mann–Whitney *U*-test was used to analyse the overall difference between groups. Differences were considered significant when *P*-value < 0.05. All data are rapresented as mean ± SEM. Source data are provided as a Source Data file.

of FMD for anti-PD1-antiCTLA4 and two cycles of FMD for anti-OX40-anti-PD-L1, as shown in scheme of Figs. 1A, B) (Fig. 5A, B).

### FMD reduces cardiac and systemic pro-inflammatory cytokines in melanoma-bearing mice treated with combinatorial ICIs therapies

Cytokine storm in the heart during anticancer therapies is a key driver of multiple cardiotoxic events[49]. Therefore, we quantified key cytokines and growth factors in plasma and in heart tissue of anti-OX40/anti-PDL1- or anti-PD-1/anti-CTLA-4-treated mice fed with FMD or a standard diet (one cycle of FMD for anti-PD1/antiCTLA4 and two cycles of FMD for anti-OX40/anti-PD-L1, as shown in the scheme of Fig. 1A, B; analysis performed on the last day of FMD) (Fig. 6A, B). IL-1α and IL-1β in the heart increased upon anti OX40/anti-PD-L1 treatment when compared to the levels in untreated mice (anti-OX40/anti-PD-L1 132,2 ± 7,3 vs. IgG 73.2 ± 5.3 pg/mg of tissue and

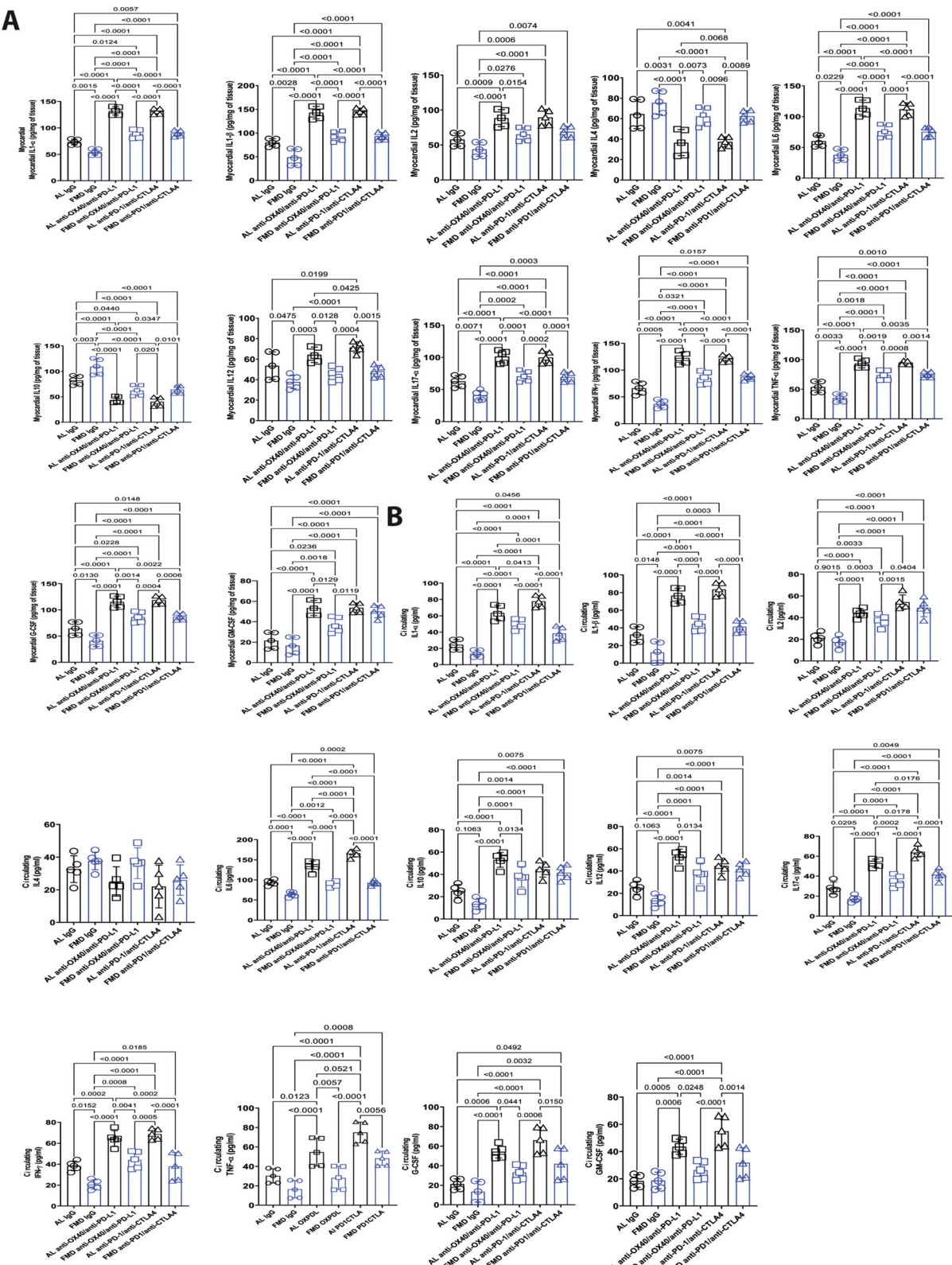

**Fig. 6 | FMD affects pro- and anti-inflammatory cytokine secretion upon immunotherapy treatment.** Cytokines level (pg/ml) in heart (**A**) and plasma (**B**). IL-1α, 1β, 2, 4, 6, 12, 17α interleukin-1α, 1β, 2, 4, 6, 12, 17α, IFN-γ interferon γ, TNF-α tumor necrosis factor α, G-CSF granulocyte colony stimulating factor, GM-CSF granulocyte-macrophage colony stimulating factor ($n$ = 5). Statistical analysis was performed using one-way analysis of variance (ANOVA). Differences were considered significant when $P < 0.05$. All data are represented as mean ± SEM. Source data are provided as a Source Data file.

anti-OX40/anti-PD-L1 143,1 ± 12.1 vs. IgG 78.4 ± 8,7 pg/mg of tissue, IL-1α and IL-1β respectively; $p < 0.01$) (Fig. 6A). A similar trend was seen for IL-6 and IL17-α, both pro-inflammatory cytokines involved in myocardial damage (anti-OX40/anti-PD-L1 112.5 ± 13.2 vs. IgG 57.8 ± 12.1 pg /mg of tissue for IL-6; anti-OX40/anti-PD-L1 98.5 ± 9.5 vs. IgG 62.1 ± 9.6 pg/mg of tissue for IL17-α; $p < 0.01$ for all). Inversely, IL-4 and IL-10, anti-inflammatory cytokines, decreased in myocardial tissue of anti OX40/anti PD-L1 treated mice compared to the untreated standard diet group ($p < 0.01$). Moreover, INF-γ, inflammation marker and a typical Th1 response mediator, was higher ($p < 0.01$) in the anti OX40/anti PD-L1 group than in the untreated mice. Notably, a highly significant increase was observed in the heart levels of TNF-α, G-CSF, and GM-CSF in the anti-OX40/anti-PD-L1 group compared to the levels observed in the control group. A similar behavior was seen in plasma concentration of cytokines and chemokines (Fig. 6B), indicating systemic pro-inflammatory effects of anti-OX-40 associated to anti-PD-1 therapy.

When mice were subjected to FMD, the levels of several cytokines and growth factors underwent strong changes in both myocardial tissue and plasma (Fig. 6A, B). FMD reduced the expression levels of cytokines IL1α, IL1β, IL-6, il-17α, INF-γ, TNF-α, and of factors G-CSF and GM-CSF in the heart and plasma of the untreated or anti-OX40/anti-PD-L1 or anti-PDL-1/anti-CTLA4 treated mice, while increasing or not affecting the release of anti-inflammatory cytokines such as IL4 and IL10 (Fig. 6A, B).

### FMD reduces systemic levels of hydrogen peroxide and leukotrienes as well as the myocardial expression of NLRP-3, leukotrienes and NF-kB in B16-F10 melanoma-bearing mice treated with combinatorial ICIs therapies

We investigated the cardiac markers of inflammation in mice fed with a standard diet or FMD, and treated with anti-OX40/anti-PDL-1 or anti-PD-1/anti-CTLA-4. In the standard diet group + immunotherapy, the levels of hydrogen peroxide, a reactive oxygen species (ROS), in heart lysates increased by approximately three fold compared to that in the untreated group (AL anti-OX40/anti-PD-L1 3.75 ± 0.2, vs AL anti-PD-1/antiCTLA-4 4.21 ± 0.3 vs. AL IgG 1.34 ± 0.17 nmol/mg of tissue; $p < 0.01$) (Fig. 7A) indicating a high degree of oxidative stress and damage in the heart during ICIs. A similar change was seen for NLRP3, an inflammatory marker, (8.7 ± 0.7, vs 9.15 ± 0.9 vs. 2.8 ng/mL ± 0.6 for anti-OX-40/anti-PD-L1, anti-PD-1/anti-CTLA-4 and the control group, respectively; $p < 0.01$), leukotrienes, lipid mediators involved in acute and chronic inflammation (68.6 ± 3.7 vs. 83.3 ± 3.1 vs 33.5 pg/mL ± 3.2 for anti-OX-40/anti-PD-L1, anti-PD-1/anti-CTLA-4 and the control group, respectively; $p < 0.01$;) and NF-kB, which regulates the expression of pro-inflammatory cytokine genes and inflammasome activity (32.8 ± 4.3 vs. 42.2 ± 2.1 vs 18.8 pg/mL ± 3.8 for anti-OX-40/anti-PDL1, anti-PD-1/anti-CTLA-4 and the control group, respectively; $p < 0.01$) (Fig. 7A).

Myocardial levels of hydrogen peroxide, NLRP3, leukotrienes and NF-kB were reduced in FMD groups alone or combined to combinatorial ICIs therapies (OX40/PDL-1 and PD-1/CTLA-4) indicating antioxidant and anti-inflammatory properties for FMD treatment (Fig. 7A).

A similar effect was seen for systemic biomarkers of inflammation and oxidative stress (Fig. 7B). Plasma hydrogen peroxide levels increased drastically after combinatorial ICIs therapy in AL groups, indicating pro-oxidative stress induced by immunotherapy. FMD treatment prevented this increase with a return of hydrogen peroxide levels to levels comparable to those in mice not treated with ICI. Similar effects were observed for systemic leukotrienes B4 levels (pg/ml), eicosanoids involved in melanoma progression and survival (Fig. 7B). Together, these results indicate that ICI triggers an inflammatory response and damage in the heart which involves hydrogen peroxide and leukotrienes, both of which are reduced by FMD treatment.

### FMD cycles cause a non significant trend for improved immunotherapy efficacy against LLC1 lung carcinoma but prevents immune related cardiac adverse events

Ajona et al. showed that three cycles of fasting occurring every three days in combination with anti-PD-1 ICI were so effective in improving the effect of ICI in delaying tumor progression to not only cause a slow down but a regression of LLC tumors[36]. Here, because our focus was on side effects and not cancer progression we tested the effect of only two cycles of FMD occurring every week in combination with immunotherapy. The anti-PD-1/anti-CTL4 combination was administered simultaneously, while anti-OX40/anti-PD-L1 were administered sequentially (anti-OX40 the first week and anti-PD-L1 the second week) to avoid that the combination could cause T cell apoptosis and exhaustion as reported in the literature (Fig. 8A). Anti PD-1/anti-CTLA4 therapy either with or without FMD (two cycles) show no benefit against LLC1 tumor growth, whereas FMD in combination with anti-OX40/anti-PD-L1 treatment was the most effective in reducing tumor masses, but this effect was not statistically better than anti-OX40/anti-PD-L1 alone (Fig. 8B). The analysis of the immune infiltrate showed that anti-OX40/anti-PD-L1 treatment increases the percentage of CD8 and CD4 (Fig. 8C, E) and reduces the percentage of Treg (Fig. 8F), while anti-PD1/anti-CTLA4 has no effect on the CD4 and CD8 immune infiltrate and on Treg cells compared to the untreated group (Fig. 8C, E, F). Although FMD did not modify the percentage of immune infiltrated CD8 and CD4 T cells in the various experimental groups, the IHC analysis of the tumor sections showed that FMD only in combination with anti-OX40/anti-PD-L1 promotes CD8 T cells infiltration into the tumor center (Supplementary Fig. 4A, B), but it does not affect CD4 T cells (Supplementary Fig. 4C).

We did not observe statistically significant differences related to the CD8 activation status (GZMB+ and KI67+) both between the standard diet and FMD groups and between the untreated and treated groups (Fig. 8D, Supplementary Fig. 5A). Notably, FMD reduced the percentage of CD25+ Tregs in combination with both immunotherapies (Supplementary Fig. 5B) and the percentage of CTLA4+ Tregs in combination with anti-OX40/anti-PD-L1 (Supplementary Fig. 5C).

Regarding the myeloid compartment, the anti-OX40/anti-PD-L1 combination reduced the percentage of macrophages (F480+) (Supplementary Fig. 5D), but FMD did not affect either the macrophage population or their polarization state (CD206+F4/80+) (Supplementary Fig. 5D, E).

There were no significant changes in the levels of PMN-MDSC (GR1high) and PMN-MDSC with suppressive phenotype (PD-L1+ GR1high) between the various experimental groups (Supplementary Fig. 5F, G; Supplementary Fig. 4E), but FMD reduced the percentage of M-MDSCs (LY6Chigh) (Supplementary Fig. 5H) and increased the percentage of M-MDSCs with suppressive phenotype (PD-L1+LY6high) in combination with anti-OX40/anti-PD-L1 (Supplementary Fig. 5H).

In contrast, myocardial analysis in heart tissue of mice has shown an anti-inflammatory phenotype induced by FMD in lung cancer bearing mice treated with combinatorial ICIs therapies (Fig. 8G–M). First, immunohistochemical and biochemical analysis indicates that mice treated with anti-OX40/anti-PD-L1 and anti-PD1/anti-CTLA4 monoclonal antibodies experienced myocardial damages, including a slight increase in fibrosis, hypertrophy and necrosis compared to those in the control group (IgG treated mice) (Fugure8 G, I). Similarly, higher myocardial CD3+ and CD8+ lymphocyte staining was seen in AL anti-OX40/anti-PD-L1 and anti-PD1/anti-CTLA4 groups compared to the IgG control groups (Fig. 8H, M).

FMD cycles also caused a reduced frequency of necrosis and absence of fibrosis (Fig. 8G, I) as well as reduced CD3+ and CD8 + lymphocyte staining in myocardial tissue compared to those in AL mice treated with ICIs (Fig. 8H, M), as supported by both immunohistochemical staining and quantitative data of CD3+ and CD8+ lymphocyte counts /mm².

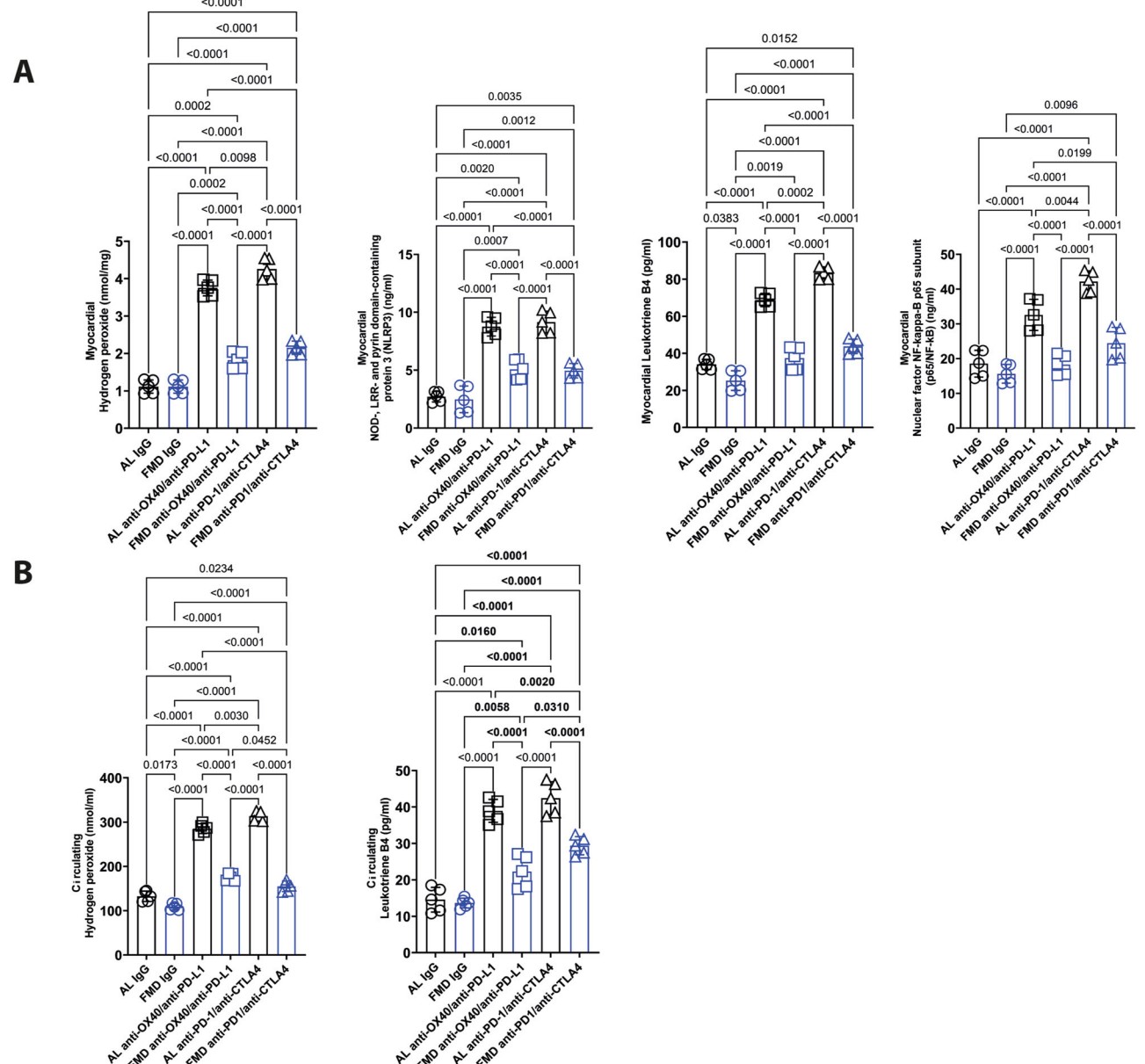

**Fig. 7 | FMD exerts cardio-protection against immunotherapy side effects via attenuating Reactive Oxygen Species (ROS), NLRP-3 inflammasome, leukotrienes and NF-kB expression in heart.** ROS, NLRP-3 inflammasome, leukotrienes and NF-kB expression in heart (**A**) and plasma (**B**) (*n* = 5). Statistical analysis was performed using one-way analysis of variance (ANOVA). Differences were considered significant when *P* < 0.05. All data are represented as mean ± SEM. Source data are provided as a Source Data file.

Expression of mouse pro-collagen 1α1 and metalloproteases type 9 (MMP-9), biomarkers of fibrosis and inflammation, respectively, was strongly increased in AL anti-OX40/anti-PD-L1 and anti-PD1/anti-CTLA-4 groups compared to those in AL IgG mice, an effect previously described[50] (Fig. 8L). FMD reduced pro-collagen 1α1 and MMP-9, corroborating those of IHC indicating an anti-fibrotic and anti-inflammatory effect (Fig. 8I).

Inflammasome NLRP-3, is a key trigger of heart failure, arrhythmias, myocarditis and cancer progression. We investigated whether FMD could affect the NLRP-3 inflammasome, leukotrienes, NF-kB and hydrogen peroxide levels in myocardial tissue during ICIs therapies (Supplementary Fig 6C). In AL anti-OX40/anti-PD-L1 and anti-PD1/anti-CTLA4 groups, hydrogen peroxide, NLRP-3 inflammasome, leukotrienes and NF-kB myocardial expression were increased, compared to those in the IgG control group (p < 0.001), indicating the induction of a pro-inflammatory phenotype induced by ICIs therapies (Suppl. Figure 6C, D). Notably, FMD reduced all analyzed inflammatory biomarkers, confirming its anti-inflammatory properties in myocardial tissue (Suppl. Figure 6C, D). Because, NLRP-3 induces hypersecretion of several cytokines involved in cardiovascular diseases and cancer, we analyzed systemic and myocardial cytokines and growth factors levels. In brief, mice on the AL diet and anti-OX40/anti-PD-L1 or anti-PD1/anti-CTLA-4 therapies experienced high IL-1α, IL-1β, IL-6, IL-17 α, INF-γ, TNF- α, and GM-CSF levels compared to control. FMD caused a reduction in both systemic and myocardial expression of cytokines (Supplementary Fig 6A, B).

Taken together, these results indicates that in mice ICIs therapies cause a wide range of strong inflammatory and oxidative stress responses which damage cardiac tissue. FMD cycles were able to prevent or reverse the great majority of these damaging responses.

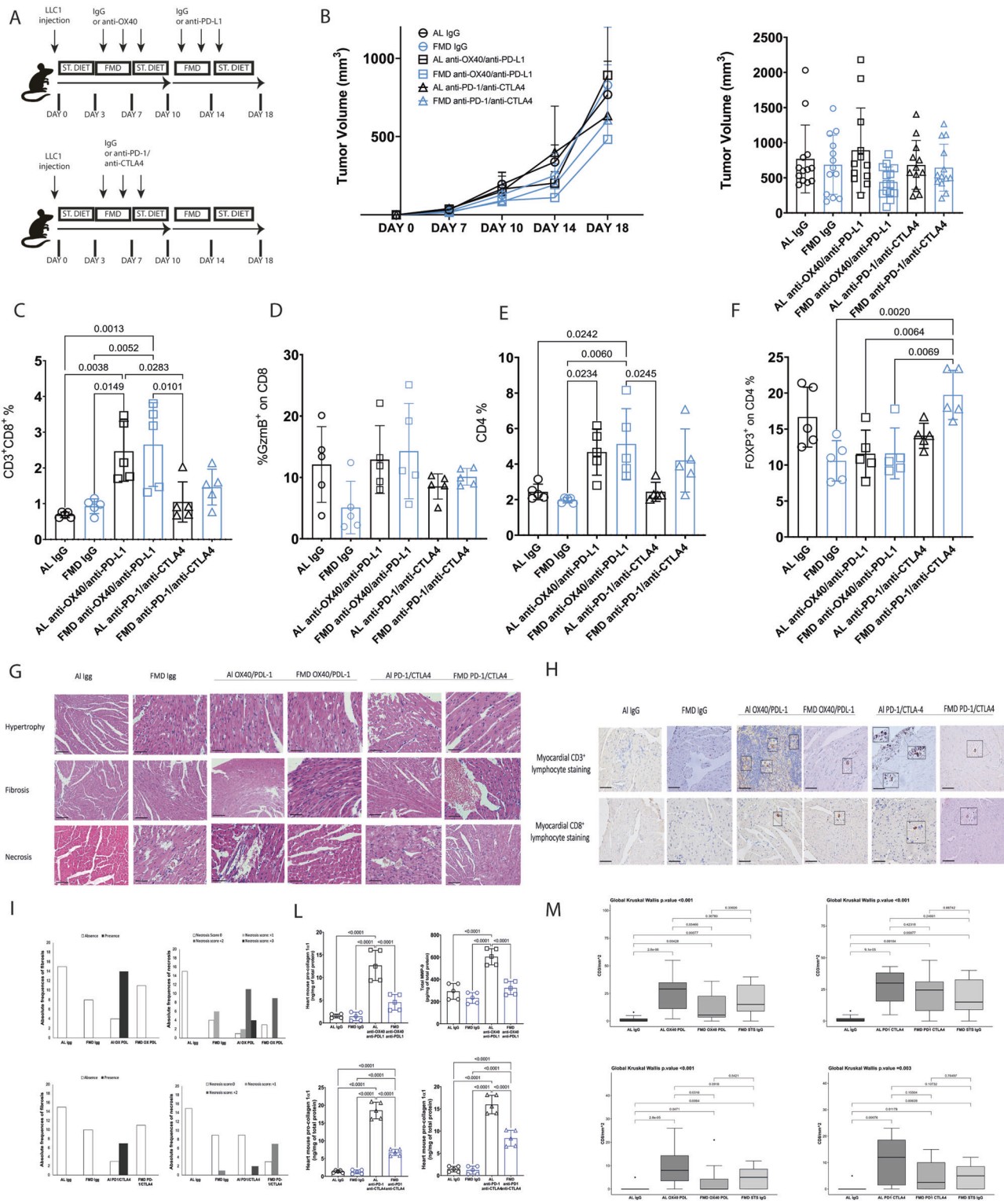

## Discussion

In this study we show that FMD cycles alone delay melanoma but not lung cancer growth and show that FMD protects the heart from the adverse effects of immune therapy (anti-OX40/anti-PD-L1, anti-PD1/anti-CTLA4), by reducing autoimmune responses, lymphocyte infiltration and preventing inflammatory and oxidative damage in heart tissue in both the B16-F10 melanoma and LLC1 lung cancer models.

As suggested by previous studies, our results here indicate that multiple and possibly many FMD cycles are required to cause strong anti-cancer effects. In fact, two FMD cycles were the minimum required to cause a significant delay in melanoma growth, although even two FMD cycles did not cause significant effects in the lung cancer model. Notably, FMD alone is much more effective in controlling the growth of melanoma than immunotherapy alone and even for lung cancer, both our study and a published study[36] indicate that, without fasting/FMD, immunotherapy is not effective against a lung tumor. Similarly, for breast cancer cells, the FMD alone blocks the proliferation of 4T1 breast cancer, a glycolysis-dependent tumor line, while it has no effect on TS/A breast cancer growth, whose metabolism is based on

**Fig. 8 | FMD does not improve immune-therapy efficacy against lung LLC1 cancer, but reduces immune-therapy related cardiac damage. A** Schedule of tumor implantation and treatment for B16F10 syngeneic tumor models. **B** LLC1 Tumor growth in immunocompetent C57/BL6 syngeneic mice treated with isotype control, anti-OX40/anti-PD-L1 and anti-PD-1/anti−CTLA4 and fed with standard diet or FMD (AL IgG *n* = 13; FMD IgG *n* = 13; AL OX40PDL1 *n* = 12; FMD OX40PDL1 *n* = 14; AL PD1CTL4 *n* = 12; FMD PD1CTLA4 *n* = 13). Analysis of tumor immune infiltrate by FACS: **C** CD8⁺ on CD45⁺ T cells (*n* = 5); **D** GZMB⁺ on CD8⁺ T cells (*n* = 5); **E** CD4⁺ on CD45⁺ T cell (*n* = 5); **F** FOXP3⁺ on CD4⁺ T cells (*n* = 5); **G** H&E staining in the heart of C57/J mice bearing LLC1 lung tumor and treated with IgG or anti-OX40/anti-PD-L1 or anti-PD-1/anti-CTLA-4; **H** CD3 and CD8 staining in the heart of C57/J mice bearing LLC1 lung tumor and treated with IgG or anti-OX40/anti-PD-L1 or anti-PD-1/anti-CTLA-4 (3 tissue sections for each tumor; 5 tumors per each group); **I** Quantification of heart fibrosis and necrosis in the different experimental group; **L** Quantification of pro collagen 1α1 and MMP9 in the heart belonging to different experimental group (*n* = 5); **M** Quantification of CD3 and CD8 cell count in heart belonging to different experimental group (3 tissue sections for each tumor; 5 tumors per each group). Statistical analysis was performed using one-way analysis of variance (ANOVA) and Global Kruskal Wallis test. *P* values were determined by One-way ANOVA with Tukey's post analysis. Differences were considered significant when *P* < 0.05. All data are represented as mean ± SEM. Source data are provided as a Source Data file.

OXPHOS, although FMD cycles are effective against both breast cancer models when combined with ICIs[37].

It is worth noting that even if FMD does not improve the efficacy of combined anti-PD-1/anti-CTLA-4 therapy against melanoma and lung cancer, FMD in combination with anti-OX40/anti-PD-L1 causes a strong delay in cancer growth in a sub-group of the animals tested indicating that, as observed in the clinical trials with melanoma patients, immunotherapy efficacy is likely affected by both characteristics of the tumor and the host.

Since we did not observe major differences in T cell activation and myeloid population between FMD and standard diet + immunotherapy in melanoma and lung cancer, the anti-tumoral effect exerted by FMD on tumor growth could be due more to inhibition of nutrient signaling pathway and to metabolic changes which challenge the ability of tumor cells to adapt to very hostile environment lacking in nutrients and growth factors essential to sustain cell proliferation, as previously demonstrated[31,33,35,37].

FMD can reduce pro-oxidative stress and NLRP3-derived inflammation in myocardial tissues of melanoma-bearing mice, indicating cardioprotective effects. Pro-inflammatory eicosanoids, including leukotrienes B4, are involved in asthma, cancer progression and survival, psoriasis, mucositis and cardiovascular diseases[51]. Here, reduction of leukotrienes B4 is achieved by FMD cycles. Notably, fasting can reduce leukotriene formation from neutrophils in patients with rheumatoid arthritis, together with an altered fatty acid composition of membrane phospholipids[52]. Other preclinical studies demonstrated that fasting reduces linoleic acid desaturase activity, resulting in reduced arachidonic acid and pro-inflammatory eicosanoids involved in atherosclerosis and cardiovascular diseases[53]. In line with the literature, we showed that FMD can also decrease systemic leukotrienes B4 levels also during combinatorial ICIs therapies.

The side effects of ICIs are an essential point to consider in the clinical management of cancer patients with the most frequent adverse events including rash (maculopapular, lichenoid), pruritus, vitiligo, diarrhea, colitis, lichenoid mucositis, hypothyroidism, hyperthyroidism, thyroiditis, hypophysitis, transaminitis, hepatitis, pneumonitis, inflammatory arthritis, nephritis[54]. The combination of an anti-CTLA-4 and anti-PD-1 immunotherapy can cause a major increase in the portion of severe immune related side effects consistently causing grade 3 or above adverse events in over 50% of patients[20]. Myocarditis, was shown to affect 1.14% of the cancer patients receiving immunotherapy drugs but it could be much higher in patients receiving specific combination of ICI drugs. Although rare, ICI cardiotoxicity is a severe side effect that results in cardiomyopathy, myocarditis, and pericarditis, but more minor and less detectable cardiotoxicities could be much more common. A recent World Health Organization report (VigiBase) stated that ICI treatment led to an 11-fold increase in the incidence in myocarditis, with an extremely high fatality rate of 46% in combinatorial therapies compared to monotherapies. Nivolumab (anti-PD1) and Ipilimumab (anti-CTLA4) are associated with cardiovascular injuries, especially when administered in combination[55,56].

Notably, to date, the mechanisms and key players of ICIs-induced myocardial injuries are not entirely understood. Immune cell uptake and infiltration in myocardial tissue are seen in human histological studies with high amounts of CD4⁺/CD8⁺ T lymphocytes and macrophages (CD68⁺ cells) that involve chemokines including CXCR3, 9, and 10, which increase granzyme B-mediated cytotoxicity, driving cardiac injury.

Among co-stimulatory targets, the OX40 (also known as CD134), a co-stimulatory molecule transiently expressed on activated human T cells, functions in T-cell activation, expansion, differentiation, generation, and maintenance of memory T cells are of particular clinical interest in cancer immunotherapy. Several agonistic anti-OX40 antibodies are currently being evaluated in phase I/II clinical trials either as monotherapies or combined with other immunotherapies in patients with malignant tumors. To date, agonistic OX40 monotherapy has led to tumor regression or stable disease in patients with solid tumors but clinical data for the combination with a checkpoint blockade are still not available[57–60].

As new clinical trials are testing whether hypoglycemia or new antidiabetic drugs can increase immune function through AMPK-mTOR-related pathways and protect against chemotherapy cardiotoxicity, our results raise the possibility that FMD cycles by reducing glucose concentration and activating AMPK inhibit the activation of NLRP3 and NF-KB, responsible for the release of pro-inflammatory interleukins. Fasting/FMD are also known to cause a transient increase in cortisol[61] which could regulate some of the anti-inflammatory responses in response to immunotherapy.

Furthermore, FMD may reduce the production of ROS caused by ICI treatment in the heart but not cancer cells, and therefore reduce the production of leukotriene and pro-inflammatory factors through the inhibition of NLRP3 pathway by inducing a differential anti-stress response in normal cells[32] (Supplementary Fig. 7).

Interestingly, FMD also changes systemic and cardiac cytokines and growth factors involved in cancer and cardiovascular diseases-induced by immune checkpoint inhibitors (such as myocarditis, vasculitis and Takotsubo syndrome)[62,63]. These FMD-induced changes in systemic and cardiac cytokines could also affect the onset of myocarditis seen in patients with Covid 19 or vaccinated with RNA. Although the risk of developing myocarditis is rare both in patients with Covid 19 and in vaccinated people[64,65], this condition is associated with systemic inflammation and cytokine storm[66] that determines interstitial cardiac infiltration of macrophages or multifocal lymphocytes[67]. A key factor for the cytokine storm is Interleukin 6 (IL-6) which causes the activation of the T lymphocyte and the release of inflammatory cytokines, generating a positive feedback of immune activation that leads to myocardial damage[68].

This study sets the stage for clinical trials aimed at assessing the ability of FMD to increase the efficacy of immunotherapy while reducing its side effects. These results also indicate that the anti-inflammatory and protective effects of FMD cycles in combination with ICI could affect other organs and systems.

## Methods
Our studies were carried out in compliance with all relevant ethical regulations. The in vivo experiments on animals were performed

according to the protocols approved by the IFOM Institutional Animal Care and Use Committee (IACUC) and the Ministry of Health.

## Tumor cell lines

The B16-F10 melanoma tumor cell and LLC1 cell were obtained from ATCC. Cells were grown in DMEM supplemented with 10% fetal bovine serum (FBS), 2 mM l-glutamine, penicillin (100 U ml$^{-1}$) and strepto-mycin (100 μg ml$^{-1}$) at 37 °C with 5% $CO_2$ and maintained at a con-fluence of 70–80%.

## Tumor implantation, Immune checkpoint blockade (ICB) treatment and tumor volume measurement

C57BL/6J female mice, 6–8 weeks old, were purchased from Charles River and housed under pathogen-free conditions at $22 \pm 2$ °C with $55 \pm 10\%$ relative humidity and with 12 h day/light cycles in Cogentech animal facility and with food and water ad libitum. All procedures were carried out in accordance with the guidelines established in the Prin-ciples of Laboratory Animal Care (directive 86/609/EEC), were approved by the Italian Ministry of Health, and were performed under the supervision of Institutional Animal Care and Use Committee (OPBA) at IFOM- The AIRC Institute of Molecular Oncology.

In the vivo experiments, tumors were implanted in C57BL/6J mice by injecting subcutaneously (s.c.) $2 \times 10^5$ B16-F10 or $5 \times 10^5$ LLC1 cells per mouse into the right flank at day 0. Three days after tumor injection, mice from the appropriate groups (at least 5 mice per group) were treated intraperitoneally (i.p.) with anti-PD-L1 (at the dose of 100μg per mouse), anti-OX40 (at dose of 100 μg per mouse), anti-PD1 (at the dose of 100μg per mouse), anti-CTLA4 (at dose of 100 μg per mouse). The ICB therapy was administered every other day for three treatments. The combined anti-OX40/anti-PD-L1 treatment was administered sequen-tially. The mice were treated with anti-OX40 the first week, while the second week with anti-PD-L1. Anti-PD-1/anti-CTLA4 were administered concurrently on 4, 6 and 8 post-injection day. The mice underwent one or two cycles of FMD (4 days each week) starting the third day after tumor implantation and sacrificed on 21 post-injection day.

One FMD cycle consists of alternating four consecutive days of fasting mimicking diet and three days of refeeding with standard diet. FMD components are described in Brandhorst et al. and Di Biase et al.[38,40]. Briefly the day 1 diet provides 1.88 kcal/g (50% of normal daily intake) and is made by a mix of various low-calorie broth powders, a vegetable medley powder, extra virgin olive oil, and essential fatty acids mixed with hydrogel; day 2-4 diet contains 0.36 kcal/g (10% of normal day intake) and consist of low-calorie broth powders and gly-cerol mixed with hydrogel.

Tumors were measured every 3–4 days using a digital caliper; tumor volume was calculated using the formula $V = (L \times W \times H)/2$, where V is tumor volume, L is the length of the tumor (longer diameter), W is the width of the tumor (shorter diameter) and H is the height (diameter of tumor perpendicular to length and width). Mice were monitored for tumor growth and survival. Mice were killed when tumor volume reached 1.5 cm$^3$.

## Flow cytometry analysis of tumor-infiltrating lymphocytes and apoptosis

For the flow cytometry analysis of tumor-infiltrating lymphocytes, tumors were minced, B16 were digested for 1 h with Collagenase D (10 mg/ml) and DNAseI (10 mg/ml). Processed tumors were load on Lympholyte gradient and centrifuged at $1500 \times g$ for 30 min. The interphase ring, which contains most live leukocytes cells, was col-lected and used for FACS staining.

$1–2 \times 10^6$ cells per sample were stained with the LIVE/DEAD stain (Invitrogen), and then with membrane protein marker (CD45, CD3, CD8, CD4, CD44) followed by fixation with formaldehyde. For intra-nuclear (Foxp3, Tbet, Klrg1, Eomes, Tcf1/7) and cytoplasmic marker (GzmB) staining, cells were permeabilized and fixed with Foxp3/

transcription factor staining kit (Invitrogen eBioscence) or BD cytofix/cyto perm kit (BD biosciences). Data acquisition was performed on Attune NxT Flow Cytometer. Results were analyzed with the FlowJo software. Gating strategies for myeloid and lymphoid cells are illu-strated in Supplementary Fig 8.

| REAGENT or RESOURCE | SOURCE | IDENTIFIER |
|---|---|---|
| **Antibodies FACS** | | |
| Anti-mouse/human CD44, PB (IM7) | Biolegend | 103020 |
| Anti- muse FoxP3, eFluor 506 (FJK-16s) | eBioscience | 69-5773-82 |
| Anti-mouse CD274 (PD-L1, B7-H1), Biotin (1-111 A) | eBioscience | 13-9971-81 |
| Anti-mouse Eomes, PE (Dan11mag) | eBioscience | 12-4875-82 |
| Anti-mouse CD127 (IL-7Rα), PE (A7R34) | Biolegend | 135010 |
| Anti-mouse CD8a, PerCP-Vio700 (53-6.7) | Miltenyi Biotec | 130-120-756 |
| anti-mouse CD185 (CXCR5), PE/Cy7 (L138D7) | Biolegend | 145516 |
| Anti-mouse CD335 (NKp46), PE-eFluor 610 (29A1.4) | eBioscience | 61-3351-82 |
| Anti-mouse CD366 (Tim-3), PE/Dazzle 594 (B8.2C12) | Biolegend | 134013 |
| Anti-mouse T-bet, AF 647 (4B10) | Biolegend | 644804 |
| Anti-mouse/human KLRG1, APC (2F1/KLRG1) | Biolegend | 138412 |
| Anti-mouse Ly108, APC (330-AJ) | Biolegend | 134610 |
| Anti-mouse CD3, AF 700 (17A2) | eBioscience | 56-0032-82 |
| Anti-mouse MRC1, AF 700 (MR6F3) | eBioscience | 56-2061-82 |
| Anti-mouse CD25, APC/Cy7 (3C7) | Biolegend | 101918 |
| Anti-mouse PD1, Biotin (RMP1-30) | Biolegend | 109106 |
| Anti-mouse CD11c, APC-Vio770 (N418) | Miltenyi Biotec | 130-107-461 |
| Anti- mouse TCF1/TCF7, AF 488 (C63D9) | Cell Signaling Technology | BK6444S |
| Anti-mouse CD11b, VioBlue (REA592) | Miltenyi Biotec | 130-113-810 |
| Anti-mouse CD4, VioBright FITC (REA604) | Miltenyi Biotec | 130-118-692 |
| Anti-mouse Granzyme B, FITC (REA226) | Miltenyi Biotec | 130-118-341 |
| Anti-mouse Ly-6C, FITC (REA796) | Miltenyi Biotec | 130-111-915 |
| Anti-mouse MHC Class II, PE (REA813) | Miltenyi Biotec | 130-112-231 |
| Anti-mouse CD62L, PE (MEL-14) | Biolegend | 104407 |
| Anti-mouse CTLA-4, BV421 (UC10-4B9) | Biolegend | *106311* |
| Anti-mouse Ly-6G, PerCP-Vio700 (REA526) | Miltenyi Biotec | 130-117-500 |
| Anti-mouse CD45, PE-Vio770 (REA737) | Miltenyi Biotec | 130-110-661 |
| Anti-mouse F4/80, APC (REA126) | Miltenyi Biotec | 130-116-525 |
| **Antibodies in vivo** | | |
| anti-mouse OX40 (CD134) | BioXcell | BE0031 |
| anti-mouse PD-L1 (B7-H1) | BioXcell | BE0101 |
| anti-mouse CTLA4 (CD152) | BioXcell | BP0032 |
| anti-mouse PD-1 | BioXcell | **BE0146** |
| **Antibodies IHC** | | |
| Anti-CD3 | Abcam | Ab16669 |
| Anti- CD8α | Abcam | Ab217344 |
| Anti-Myeloperoxidase | Abcam | Ab208670 |
| Anti-B220 | Bd bioscience | 553928 |

## Collagen quantification in myocardial tissues

For ex vivo analyses, hearts were excised and fixed in 10% neutral buffered solution. The myocardial tissues were formalin-fixed and paraffin-embedded for morphometry and immunohistochemistry.

General morphology was studied using hematoxylin-eosin staining. Sections were stained with hematoxylin/eosin to evaluate the collagen content in tissue. In order to quantify the total collagen content in the heart tissues, measurement of pro-collagen 1α1 (an established biomarker of cardiac fibrosis) was performed using the Mouse Pro-Collagen I alpha 1 CatchPoint SimpleStep ELISA Kit from Abcam (ab229425). Tissues were homogenized, after protein quantification (Bradford assay) 100 μg of proteins were assayed according to manufacturer's instruction. Fluorescence was measured at 530/590 using a 96-well plate reader Tecan Infinite M200 plate-reader (Tecan Life Sciences Home, Männedorf, Switzerland). Metalloproteases type 9 (MMP-9) were associated with collagen maturation in heart failure, demonstrating the important role of these enzymes in fibrosis through collagen configuration, activation, and deposition. Therefore, we quantified MMP-9 in heart tissues through Mouse MMP9 ELISA Kit (ab253227, Abcam, Milan, Italy) following the manufacturer's instructions; results are expressed as pg of MMP-9/mg of protein (determined through Bradford assay).

### Immunohistochemistry

Mouse tissues were fixed in 10% buffered formalin and paraffin embedded with Diapath automatic processor. To assess histological features Hematoxylin/Eosin (Diapath) staining was performed according to standard protocol and samples were mounted in Eukitt (Bio-Optica).

For immunohistochemical analysis, paraffin was removed with xylene and the sections were rehydrated in graded alcohol. Antigen retrieval was carried out using preheated target retrieval solution (pH 9.0) for 30 minutes. Tissue sections were blocked with FBS serum in PBS for 90 min and incubated overnight with primary antibodies.

For melanoma tumor antibody binding was detected using the conjugated goat anti-rabbit polymer alkaline phosphatase (AP) (Biocare) followed by a Vulcan red chromogen reaction (Peroxidase substrate kit, DAB, SK-4100; Vector Lab).

For lung tumor antibody binding was detected using a polymer detection kit (GAR-HRP, Microtech) followed by a diaminobenzidine chromogen reaction (Peroxidase substrate kit, DAB, SK-4100; Vector Lab). All sections were counterstained with Mayer's hematoxylin and visualized using a bright-field microscope (LEICA DM750)

For B220 antibody binding was detected using a ABC kit Vectastain (vector, Pk4100) followed by a diaminobenzidine or Magenta chromogen (ENvision flex hrp magenta substrate chromogen system,dako) reaction.

### Quantification of CD3⁺ and CD8⁺ cells in myocardial tissues

To evaluate for tissue abundance of $CD3^+$ and $CD8^+$ T-cells and diagnosing any myocarditis induced by immunotherapy, formalin-fixed paraffin-embedded (FFPE) sections of heart (4 μm thickness) were stained with rabbit anti mouse CD3 and CD8 antibodies, diluted 1:150 and 1:300 respectively in Dako AR9352 diluent buffer, upon antigen retrieval with 10 mM citrate buffer pH 6 (CD3) or with EDTA antigen retrieval Buffer pH 8 (CD8). Goat anti-rabbit antibodies were used as secondary antibodies for CD3 and CD8.

Visualization of the antibody–antigen reaction was visualized by peroxidase complex kit reagents (SignalStain® DAB Substrate Kit) as the chromogenic substrate. Finally, sections were weakly counterstained with hematoxylin and mounted. Lymphocyte densities (cells/mm2) were quantified by an experienced pathologist (FT).

### Quantification of plasma and myocardial cytokines

Cytokines involved in inflammation (IL-1α, IL-1β, IL-2, IL-4, IL-6, IL-10, IL-12, IL17-α, IFN-γ, TNF-α, G-CSF, and GM-CSF) were quantified in heart tissue extracts by using the 12 mouse cytokine Multiplex Assay Kit (Qiagen, Germantown, USA) following the manufacturer's instructions

and summarized in other recent work[63]. Results were expressed as pg of cytokine/mg of heart tissue.

### Quantification of pro-inflammatory biomarkers

After treatments, mice were sacrificed after the proper anesthesia as described before. Hearts were weighed and snap-frozen in dry ice; after, heart tissues were homogenized in a solution 0.1 M PBS (pH 7.4) containing 1% Triton X-100 and protease inhibitor cocktail. Tissues were well homogenized through a step in a high intensity ultrasonic liquid processor. Obtained homogenates were than centrifuged at 4 °C and supernatants were treated for quantification of several inflammation markers. Specifically, leukotriene B4 expression (pg/mL of tissue lysate) was quantified through the LTB4 ELISA Kit (Enzo, Life Technology). Nuclear factor NF-kappa-B p65 subunit (p65-NF-Kb) expression (ng/mL of tissue extract) was quantified through a mouse, rat RelA/NF-kB p65 ELISA Kit (My BioSource, Seattle, WA, USA). NLRP3 inflammasome expression (ng/mL of tissue extract) was quantified by the NLRP3 ELISA Kit (Mouse) (OKEH05486) (Aviva Systems Biology); quantification of systemic and myocardial hydrogen peroxide, a reactive oxygen species (ROS) marker, was performed by using a Fluorimetric Hydrogen Peroxide Assay Kit (Sigma Aldrich, Milan, Italy), in line with other recent work[49,69]. This kit utilizes a peroxidase substrate that generates a red fluorescent product (λex = 540/λem = 590 nm) after reaction with hydrogen peroxide that was analyzed by a fluorescent microplate reader.

### Statistical analysis

GraphPad Prism 9 was used for statistical analysis and graphing. Differences between three or more groups were analyzed using one-way ANOVA followed by post hoc analysis through Tukey's multiple comparison test. Comparison of CD3 and CD8 immune cell count in heart sections were performed using the Kruskal−Wallis test with post hoc Mann−Whitney U-test. Statistical tests are indicated in the respective sections and figure captions. Differences were considered statistically significant at p value (p) < 0.05. All data are represented as mean ± SEM of three independent experiments.

For in vivo experiments, sample size estimation was performed with G.Power software using a multifactorial variance analysis (ANOVA) repeated measurement, within-between interaction.

A power analysis with significance level $α = 0.05$, (assuming a large effect size $f = 0.4$) indicated that n = 8 mice per group are needed to achieve a power $(1 − β) = 0.9$ which is considered adequate to detect difference between means at least of two groups.

### Reporting summary

Further information on research design is available in the Nature Portfolio Reporting Summary linked to this article.

## Data availability

Data supporting this study, including Supplementary Information and Source Data, are provided with this paper article. All data generated or analysed during this study has been deposited in Figshare, https://doi.org/10.6084/m9.figshare.23283995. Source data are provided with this paper.

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

## Acknowledgements

We are thankful to IFOM and Cogentech facility for technical support. This work was supported by the Associazione Italiana per la Ricerca sul Cancro (AIRC; IG#17605 and IG#21820).

## Author contributions

S.C., V.Q., E.V., O.B., A.S., S.L. and V.S. performed all the in vivo and in vitro experiment. G.D. and C.C. carried out FACS analysis. A.D., F.T. and F.P. performed IHC analysis. G.C., P.D., C.T. and N.M. provides expertise and feedback. All authors discussed the results and contributed to the final manuscript. S.C., V.Q., and V.L. conceived and designed the study and drafted the manuscript.

## Competing interests

V.D.L. holds intellectual property rights on clinical uses of FMD and equity interest in L-Nutra, a company that markets medical food. The remaining authors declare no competing interests.
