## [Peer Review File · Nature Communications]

Fasting mimicking diet in mice delays cancer growth and reduces immunotherapy-associated cardiovascular and systemic side effectsREVIEWER COMMENTS

Reviewer #1 (Remarks to the Author):

In this manuscript, Cortellino and Quagliariello et al. used several molecular cell biology, molecular pathology and immunology experiments in mouse model to address their main research interest in investigating if FMD plays a cardioprotective role in reducing inflammatory, autoimmunity and TILs. The authors identified that (1) the efficacy of combined immune checkpoint inhibitors is independent of FMD, (2) FMD delays tumour growth, (3) modulates GZMB+ NK cells and tumour infiltrating myeloid cells while decreasing CD4+ and CD8+ T cells, and (4) reducing heart inflammation, fibrosis/necrosis and hypertrophy. Although there is a recent interest in how diet can influence modulating immunotherapy response and toxicity, this study lacks clarity and the scientific results require improvement.

Specific comments:

1. This study seems to be a follow-up work or “additional” data that is not used from their previous publication in Cell Rep 2022 Aug 23;40(8):111256. Although the authors provided good rationale in regards to their targeted panel of markers and cytokines, the sample size of this work precluded the observations and their findings in the manuscript.

1a. The first result as illustrated in Figure 1 focuses only on CD8 T and NK phenotypes. Although these are the key players in the immune microenvironment in modulating immunotherapy toxicity, there are also other immune cells such as Macrophages and B cells that can influence the treatment. Although the observation of FMD leads to the increase of TILs (T and NK cells) is interesting, the study lacks immunofluorescence data to show the infiltration of lymphocytes into the tumour.

1b. In addition, the anti-PD-L1 is stated in Figure 1, however, the result section stated anti-PD-1.

1c. The number of mice used in this study is unclear. From the given figures, it seems that the authors used an average of 5 mice in each study group for data comparison, and applied unpaired t-test for comparative analysis. The heterogeneity of data indicates that more samples are needed to increase the power of statistics so that non-parametric can be applied in this work.

2. It will be interesting to know if the effects of FMD also applied to other cancer types. For example, lung, colon and renal cancers. By generating results from other cancer types in mouse model, it will further improve the importance of this work.

3. T cell antibody panel lacks CD4 marker.

4. In their previous work (Cell Rep 2022 Aug 23;40(8):111256), the manuscript stated that “pathways enrichment analysis showed that FMD affects pathways involved in ketone body metabolism, carnitine synthesis, fatty acid oxidation, and mitochondrial electron transport chain compared with the standard diet group.” The authors should perform RNA sequencing to check if there are any changes in the regulatory/transcriptomic profiles in this study.

5. There are so many typos in this manuscript that it is quite difficult to follow how FMD associates with immunotherapeutic toxicity and not treatment responses. Perhaps the

authors can further includes mouse model of responding tumours and non-responding tumours to combinatorial therapies (anti-PD-1+anti-CTLA-4 and anti-OX40+anti-PD-L1) and assess how FMD influences responses and toxicity. Although Figure 1 briefly addressed this, the data presented is not clear as the samples are not grouped into response (shrinkage of tumour) and non-response. Also, is toxicity independent of immunotherapeutic response?

6. It will also be interesting to perform their work on patient-derived tumours to further validate their work. The results generated from patient-derived tumours will further support their findings. As reported in a plethora of literatures, the immunity/immune responses from the mouse are different from human, thus testing it in patient-derived tumours will further enhance the knowledge of how diet reduce toxicity.

Minor comments:

1. It seems that the manuscript is finished off in a rush. Some of the phrases require clarity. There are multiple grammatical and typographical errors throughout the manuscript. For example, it stated that “cancer therapeutics with the potential to increase survival.” do the authors mean “increase the overall survival outcomes of cancer patients”? Another example, “preventing the onset of unwelcome side effects.” do the authors mean “unnecessary or unwanted side effects?” Other examples, “Cd45+” in Figure 3; “Figure 6” in Figure 6; “combo therapy” in Figure 1 title; is it anti-PD-L1 or anti-PD-1 in Figure 1 (the manuscript stated combined anti-PD-1 and anti-CTLA4, however, the figure caption stated anti-PD-L1/anti-CTLA-4)?

2. The authors exaggerate the toxicity in the heart. It is indeed rare (<5% of cancer patients treated with combination immunotherapies) and does not represent an urgent clinical unmet need.

Overall comment:

If the authors can perform further experiments as suggested above, the manuscript will be suitable for Nature Comm. However, the current manuscript is suitable for Scientific Reports Nature.

Reviewer #2 (Remarks to the Author):

Dr. Cortellino and colleagues perform a study of fasting mimicking diet along with combination immunotherapy regimens in mice with the B16F10 melanoma model. This poorly immunogenic model is made slightly more sensitive to PD-1/OX40 targeted regimens, though not with PD-1/CTLA-4 blockade, with slightly more immune infiltrate (albeit with slightly inconsistent results). This approach also may provide cardioprotection, although again there are some conflicting results here too. I have the following concerns and comments.

1. In the introduction, it is stated that OX40, GITR, 4-1BB targeted treatment improves anti-tumor immunity. It should be noted that while this approach may enhance responses in pre-clinical models, none of these approaches have yet been proven in patients.

2. The introduction between lines 66-102 is poorly written and paragraphs are strung together without a coherent narrative. This portion should be rewritten.

3. It seems a bit random to change from combination PD-1/CTLA-4 blockade with 1 FMD cycle to PD-1/OX40 + 2 FMD cycles. Is there a reason for this experimental design?

4. Are there effects on the innate immune system with a single cycle of FMD?
5. Figure 4B - it appears that FMD + PD-1/CTLA-4 increases necrosis in the heart compared with PD-1/CTLA-4 blockade alone? The text states the opposite.
6. Figure 5A-B - it looks like fasting diet increased T cell infiltrations in IgG treated mice. Is this mislabeled?
7. The lack of validation in other models is a potential weakness.

Reviewer #3 (Remarks to the Author):

A preclinical study was performed using melanoma-bearing mice treated with two combinatorial ICIs therapies (anti-OX-30 40/PDL-1 or antiCTLA-4/anti-PD-1) during a standard or FMD treatment regimen. Their results indicate that FMD can reduce biomarkers involved in cardiovascular disease without interfering with ICIs therapies. The premise for using FMD in combination with ICI therapy has already been established by the Longo laboratory (Cell Reports 2022). The work presented is of great interest to the immuno oncology field.

General Comments.

1. The method for FMD is not described. Without any knowledge of the methodology, it is difficult to assess it's efficacy.
2. The study lacks generalizability. The prior study using FMD in combination used a breast cancer model. Now a melanoma model was used. FMD enhanced the anti-tumor efficacy in the breast cancer model but not in the melanoma model. A more comprehensive assessment of the ability of FMD to modulate ICI treatment is needed. Furthermore, making a conclusion in the current study based on one mouse cell line lacks rigor.

Specific Comments

1. Page 8, Lines 173-174: results are overstated; the combination of FMD and anti-OX40/anti-PDL1 did not provide a significant benefit against melanoma cell growth.
2. Page 9, Lines 211—214; results are overstated; there is not a statistical analysis of fibrosis or necrosis
3. Figure 5: the number of significant digits is not consistently used
4. Methods section: there is not an analysis to determine whether the studies were adequately powered to detect differences between groups.

Below we report the responses to the reviewers point by point.

REVIEWER COMMENTS

Reviewer #1 (Remarks to the Author):

Specific comments:

1. This study seems to be a follow-up work or “additional” data that is not used from their previous publication in Cell Rep 2022 Aug 23;40(8):111256. Although the authors provided good rationale in regards to their targeted panel of markers and cytokines, the sample size of this work precluded the observations and their findings in the manuscript.

1a. The first result as illustrated in Figure 1 focuses only on CD8 T and NK phenotypes. Although these are the key players in the immune microenvironment in modulating immunotherapy toxicity, there are also other immune cells such as Macrophages and B cells that can influence the treatment. Although the observation of FMD leads to the increase of TILs (T and NK cells) is interesting, the study lacks immunofluorescence data to show the infiltration of lymphocytes into the tumour.

Below please find all the experiments (old and new) related to infiltration of T cells, NK cells, B cells, macrophages, dendritic, and myeloid-derived suppressor cells into the tumor (B16F10melanoma and LLC1 lung tumor).

The analysis of the immune infiltrate of B16 melanoma shows that immunotherapy increases CD4 immune infiltration (Supplementary Figure 3A), promotes CD8 activation (Figure 1D, Figure 2E), whereas the Treg population is reduced in the anti-OX40/anti-PD-L1 group (Supplementary Figure 3B) and increased in anti-PD-1/anti-CTLA-4 group (Supplementary Figure 1B), compared to the untreated control groups. IHC analysis of B16 melanoma tumor sections found that immunotherapy increases the percentage of CD8 in B16 melanoma tumors compared to untreated AL and FMD control groups (Supplementary Figure 2 A, B). However, in agreement with the tumor growth effects, FACS and IHC analysis showed no significant differences in the immune infiltrate between the FMD and AL groups, treated with the different combinations of immunotherapy. (Figure 1 D-F, Figure 2 D-F, Supplementary Figure 1 A-H, Supplementary Figure 2A-E, Supplementary Figure 3A-E).

FACS and IHC analysis showed no significant difference in Myeloid and B cell populations between the different experimental groups as reported in the paper supplementary figures 1-6.

One cycle of FMD, with or without anti-PD-1/anti-CTLA-4 treatment, has no effect on innate immune cell population of the melanoma TME. In fact, the immune infiltrate FACS analysis did not reveal significant differences in the M-MDSC (Supplementary Figure 1C), PMN-MDSC (Supplementary Figure 1D), dendritic cells (Supplementary Figure 1E), TAMs population (Supplementary Figure 1F) and M1-M2 TAM polarization (Supplementary Figure 1G, H), both between the various experimental groups and between the standard diet and FMD groups.

IHC myeloperoxidase staining of melanoma tumor sections showed no differences in both the number and distribution of myeloid cells within the tumor between the different experimental groups and between the standard and FMD diet groups (Supplementary Figure 2E).

Finally we investigated whether FMD or immunotherapy could affect the B-cell population within the TME. IHC staining of melanoma tumor sections with anti-B220 showed that the B cell infiltrate is very low in the tumor tissue and that in any case it is not affected by one cycle of FMD or anti-PD-1/anti-CTLA-4 immunotherapy (Supplementary Figure 2D).

On the other hand, 2 cycles of FMD decrease the percentage of M-MDSCs (Figure 3C) and increase the percentage of dendritic cells and macrophages (Figure 3A, B), but have no effect on PMN-MDSC (Figure 3D) and macrophage polarization (Supplementary Figure 3D, E). anti-OX40/anti-PDL1 reduces M-MDSCs (Figure 3C), dendritic cells (Figure 3A) and macrophages (Figure 3B) without affecting the polarization state of macrophages (Supplementary Figure 3D, E). However, staining of tumors with anti-myeloperoxidase did not show significant differences regarding the distribution of myeloid cells within the tumor tissue (Supplementary Figure 2E). The B-cell population (B220+) also did not differ in B220-stained tumor sections between the various experimental groups (Supplementary Figure 2D).

Figure 1. FMD does not improve the efficacy of anti-PD-1/anti-CTLA4 combo therapy against B16 F10 melanoma tumor. **(A)** Schedule of tumor implantation and treatment for B16 F10 syngeneic tumor models. **(B-C)** B16 Tumor growth in immunocompetent C57/BL6 syngeneic mice treated with isotype control and anti-PD-1/anti-CTLA4 and fed with standard diet or FMD. Analysis of tumor immune infiltrate by FACS:**(D)** CD8⁺ CD44⁺ GzmB⁺ cytotoxic T cell; **(E)** CD8⁺ CD44⁺ CD62L⁻ effectro memory T cell; **(F)** CD45⁺ NKp46⁺ GzmB⁺ NK cell. Statistical analysis was performed using one-way analysis of variance (ANOVA). P values : *<0,05, **<0,005, ***<0,0005.

Figure 2. FMD delays B16 tumor growth and activates NK cells. (A) Schedule of tumor implantation and treatment for B16F10 syngeneic tumor models. (B-C) B16 Tumor growth in immunocompetent C57/j syngeneic mice treated with isotype control and anti-OX40/anti-PD-L1 and fed with standard diet or FMD. (D) CD45⁺CD3⁺ T cell, (E) CD3⁺CD8⁺GzmB⁺ cytotoxic effector memory T cells, (F) CD45⁺NKp46⁺GzmB⁺ NK cells. Statistical analysis was performed using one-way analysis of variance (ANOVA). Differences were considered significant when $P \leq 0.05$. P values: * <0.05 , ** <0.005 , *** <0.0005 , **** <0.00005 .

Suppl Figure 1. Analysis of B16 breast tumor infiltrating lymphocytes upon immunotherapy treatment in standard diet and FMD group. (A) CD45⁺CD3⁺CD4⁺T cell, (B) CD3⁺CD4⁺CD25⁺Treg cells, (C) CD45⁺CD11b⁺Ly6C^{high} M-MDSC; ((D) CD45⁺CD11b⁺Ly6C^{low} Ly6G^{high}PMN-MDSC, (E) Cd45⁺ CD11c⁺MHCII⁺ dendritic cell, (F)CD45⁺ CD11b⁺F4/80^{high} macrophage, (G) CD45⁺CD11b⁺F480^{high}CD11c⁺ M1, (H) CD45⁺CD11b⁺F480^{high}MRC1⁺ M2 macrophage. Statistical analysis was performed using one-way analysis of variance (ANOVA). Differences were considered significant when P≤0.05. P values : *<0,05, **<0,005, ***<0,0005.

Supplementary Figure 2. IHC analysis of immuneinfiltrate in B16 melanoma tumor section. Tumor section stained with: A) CD8, C) CD4, D) B220, E) Myeloperoxidase. The images were obtained under 20 \times magnification. The scale bar was 100 μ m. . B) Quantification of CD8 T cells infiltration in tumor sections. Statistical analysis was performed using one-way analysis of variance (ANOVA). Differences were considered significant when $P \leq 0.05$. P values : * $<0,05$, ** $<0,005$, *** $<0,0005$.

Suppl Figure 3. Analysis of B16 breast tumor infiltrating lymphocytes upon immunotherapy treatment in standard diet and FMD group.

(A) CD45⁺CD3⁺CD4⁺T cell, (B) CD3⁺CD4⁺CD25⁺Treg cells, (C) mean fluorescence intensity (MFI) of CD127 in CD3⁺CD8⁺CD44⁺ effector T cells, (D) CD45⁺CD11b⁺F480^{high}CD11c⁺, (E) CD45⁺CD11b⁺F480^{high}MRC1⁺. Statistical analysis was performed using one-way analysis of variance (ANOVA). Differences were considered significant when P≤0.05. P values : *<0,05, **<0,005, ***<0,0005.

We then tested the effects of FMD on LLC1 lung tumor, another cold tumor insensitive to immunotherapy. Anti-OX40/anti-PD-L1 increases the percentage of CD8 and CD4 T cells in the immune infiltrate (Figure 8C, E), but has no effect on their activation status (Figure 8D), as well as on the percentage of Treg cells (Figure 8F). Although FACS analysis revealed no significant difference in CD8 T cells between the FMD and the standard diet group, the IHC-stained tumor section shows increased infiltration of CD8 cells in the tumor center of the anti-OX40/anti-PD-L1 FMD group (Supplementary Figure 4A, B). Treatment with anti-PD1/anti-CTLA-4 has no effect on tumor growth (Figure 8B), nor on the immune infiltrate of CD8 and CD4 and Tregs (Figure 4A, B). Two cycles of FMD reduce the percentage of M-MDSCs (Supplementary Figure 5H) also in the LLC1 TME, while they have no effect on the PMN-MDSC (Supplementary Figure 5F), macrophages population and their polarization status (Supplementary Figure 5D, E). In fact, no differences in the myeloid population are noted even in the sections of tumors stained with anti-myeloperoxidase (Supplementary Figure 4E). Administration of anti-OX40/anti-PD-L1 reduces macrophage infiltration (Supplementary Figure 5D), but has no effect on the M-MDSC and PMN-MDSC population (Supplementary Figure 5H, F). However FMD in combination with anti-OX40/anti-PD-L1 did not affect macrophage population in TME compared to the corresponding standard diet group (Supplementary Figure 5D). The B cell population did not vary in TME between the various experimental groups as evidenced in the anti-B220 stained tumor sections (Supplementary Figure 4D).

Figure 8. FMD does not improve immunotherapy efficacy against lung LLC1 cancer, but reduces immunotherapy related cardiac damage. **A)** Schedule of tumor implantation and treatment for B16F10 syngeneic tumor models. **B)** LLC1 Tumor growth in immunocompetent C57/BL6 syngeneic mice treated with isotype control, anti-OX40/anti-PD-L1 and anti-PD-1/anti-CTLA4 and fed with standard diet or FMD. Analysis of tumor immune infiltrate by FACS: **C)** CD3⁺CD8⁺ T cells; **D)** GZMB⁺ on CD8⁺ T cells; **E)** CD4⁺ on CD45⁺ T cell; **F)** FOXP3⁺ on CD4⁺ T cells; **G)** H&E staining in the heart of C57/J mice bearing LLC1 lung tumor and treated with IgG or anti-OX40/anti-PD-L1 or anti-PD-1/anti-CTLA-4; **H)** CD3 and CD8 staining in the heart of C57/J mice bearing LLC1 lung tumor and treated with IgG or anti-OX40/anti-PD-L1 or anti-PD-1/anti-CTLA-4; **I)** Quantification of heart fibrosis and necrosis in the different experimental group; **L)** Quantification of pro collagen 1 α 1 and MMP9 in the heart belonging to different experimental group; **M)** Quantification of CD3 and CD8 cell count in heart belonging to different experimental group. Differences were considered significant when P<0.05.

Supplementary Figure 4. IHC analysis of immuneinfiltrate in LLC1 lung tumor section. Tumor section stained with: A) CD8, C) CD4, D) B220, E) Myeloperoxidase. The images were obtained under 20 \times magnification. The scale bar was 100 μ m. B) Quantification of CD8 T cells infiltration in tumor sections. Statistical analysis was performed using one-way analysis of variance (ANOVA). Differences were considered significant when $P \leq 0.05$. P values: * <0.05 , ** <0.005 , *** <0.0005 .

Supplementary Figure 5. Analysis of LLC1 lung tumor infiltrating lymphocytes upon immunotherapy treatment in standard diet and FMD group. A) Ki67⁺ on CD8⁺ T cells; B) CD25⁺ on Treg cells; C) CTLA-4⁺ on Treg cells; D) CD45⁺ CD11b⁺F4/80^{high} macrophage; E) CD206⁺ on CD45⁺ CD11b⁺F4/80^{high} macrophage; F) CD45⁺CD11b⁺GR1^{high} PMN-MDSC G) PD-L1⁺ on GR1^{high} PMN-MDSC; H) CD45⁺CD11b⁺Ly6C^{high} M-MDSC; I) PD-L1⁺ on Ly6C^{high} M-MDSC. Statistical analysis was performed using one-way analysis of variance (ANOVA). Differences were considered significant when $P \leq 0.05$. P values: * <0.05 , ** <0.005 , *** <0.0005 .

Overall, these data, taken together, indicate that FMD in combination with OX40/PDL-1 immunotherapy is effective in causing a strong delay in melanoma growth although after 2 cycles this effect only shows a trend for improvements compared to FMD alone. Based on our previous results with a range of different tumors it is possible that additional FMD cycles would have caused a significant improvement compared to FMD alone. Similarly, for lung cancer we see only a trend for delayed tumor growth in FMD plus OX40/PDL-1 compared to OX40/PDL-1 alone. Because Ajona et al had shown synergistic effects of PDL-1 plus rapid fasting cycles against the same LLC1 lung cancer model, we believe that additional and more frequent FMD cycles would result in similar effects particularly in combination with OX40/PDL-1. However, because this has been already published we decided to focus on the effect of FMD on immunotherapy cardiotoxicity.

1b. In addition, the anti-PD-L1 is stated in Figure 1, however, the result section stated anti-PD-1.

We corrected anti-PD-L1 with anti-PD-1 in Figure 1.

1c. The number of mice used in this study is unclear. From the given figures, it seems that the authors used an average of 5 mice in each study group for data comparison, and applied unpaired t-test for comparative analysis. The heterogeneity of data indicates that more samples are needed to increase the power of statistics so that non-parametric can be applied in this work.

In this study we used at least 5 mice per group. In some graph, less than 5 samples were reported for specific staining because there were technical problems with the single sample. The immune infiltrate FACS analysis were analyzed with ANOVA and not with the student's t-test.

Although the use of additional mice would have been warranted to better separate the effects of FMD alone, immunotherapy alone or both on cancer progression, because of the strict restrictions on mouse number imposed by the Italian ministry of health, because of the focus on immunotherapy side effects and not cancer growth but also because additional mouse studies would not be possible in the 3 month turnaround requested by the editor, we have not performed additional mouse experiments, other than the lung cancer experiment.

2. It will be interesting to know if the effects of FMD also applied to other cancer types. For example, lung, colon and renal cancers. By generating results from other cancer types in mouse model, it will further improve the importance of this work.

In order to establish whether FMD reduces immunotherapy related cardiac adverse events in other tumor types, we tested the effects of FMD and immunotherapy on LLC1 lung tumor, another tumor considered cold because it is unresponsive to immunotherapy but shown by others to be sensitive to fasting plus anti-PDL-1 immunotherapy. As observed for melanoma, FMD (2 cycles) in combination with anti-OX40/anti-PD-L1 showed a trend for reduced growth of LLC1 lung tumors, (Figure 8B), while the anti-PD-1/anti-CTLA-4 FMD combination has no effect on tumor growth (Figure 8B). However, FACS analysis of the immune infiltrate did not reveal significant differences on the percentage of CD4, Tregs and CD8 T lymphocytes, on the CD8 activation (Figure 8C-F).

However, FMD in combination with anti-OX40/anti-PD-L1 increases the percentage of CD8 in the tumor center, as detected by IHC tumor section analysis (Supplementary Figure 4A, B).

As observed for the melanoma model, FMD reduces cardiac fibrosis and necrosis in this lung cancer model (Figure 8G, I, L) by limiting the infiltration of CD3 and CD8 (Figure 8H, M) into the myocardium, proinflammatory cytokine release (Supplementary Figure 6A, B) and thus preventing inflammatory myocardial damage induced by immunotherapy. Therefore we confirm in the LLC1 lung tumor model that FMD cycles reduce the risks of myocardial inflammation and of the immune related cardiac side effects induced by immunotherapy.

A**B****C****D**
Supplementary Figure 6. FMD affects pro- and anti-inflammatory cytokine secretion upon immunotherapy treatment and attenuate Reactive Oxygen Species (ROS), NLRP-3 inflammasome, leukotrienes and NF- κ B expression in heart and plasma. Cytokines level (pg/ml) in heart (**A**) and plasma (**B**). ROS, NLRP-3 inflammasome, leukotrienes and NF- κ B expression in heart (**C**) and plasma (**D**). IL-1 α , 1 β , 2, 4, 6, 12, 17 α : interleukin -1 α , 1 β , 2, 4, 6, 12, 17 α ; IFN- γ : interferon γ ; TNF- α : tumor necrosis factor α ; G-CSF: granulocyte colony stimulating factor; GM-CSF: granulocyte-macrophage colony stimulating factor. Differences were considered significant when $P < 0.05$. P values: * <0.05 , ** <0.005 , *** <0.0005 .

3. T cell antibody panel lacks CD4 marker.

In point 1a we reported the CD4 analyses performed in the immune infiltrates of melanoma and lung tumor. Both FACS and IHC analyzes (Supplementary Figure 1-4, Figure 8) showed no significant differences in myeloid cells between the standard diet and FMD groups. However, we found that 2 cycles of FMD reduced the percentage of M-MDSCs, while 1 cycle of FMD alone had no effect on this population.

4. In their previous work (Cell Rep 2022 Aug 23;40(8):111256), the manuscript stated that “pathways enrichment analysis showed that FMD affects pathways involved in ketone body metabolism, carnitine synthesis, fatty acid oxidation, and mitochondrial electron transport chain compared with the standard diet group.” The authors should perform RNA sequencing to check if there are any changes in the regulatory/transcriptomic profiles in this study.

In our experience, RNAseq on bulk RNA extracted from the tumor mass leads to high data variability within the same experimental group. This does not allow comparing gene expression profiles between different groups and detecting statistically significant changes in signaling pathways. An alternative is single cell RNAseq, which we performed in Cortellino et al Cell Repo 2022 but the entire experiment plus bioinformatic analysis would take more than 6 months, which is beyond the time the editor has allowed us to submit a revised manuscript.

5. There are so many typos in this manuscript that it is quite difficult to follow how FMD associates with immunotherapeutic toxicity and not treatment responses. Perhaps the authors can further includes mouse model of responding tumours and non-responding tumours to combinatorial therapies (anti-PD-1+anti-CTLA-4 and anti-OX40+anti-PD-L1) and assess how FMD influences responses and toxicity. Although Figure 1 briefly addressed this, the data presented is not clear as the samples are not grouped into response (shrinkage of tumour) and non-response. Also, is toxicity independent of immunotherapeutic response?

We have tried to correct all typos. In this study, FMD plus anti-OX40/anti-PD-L1 was much more effective in delaying melanoma growth compared to anti-OX40/anti-PD-L1 although this effects was not significantly different from FMD alone. So clearly, this melanoma mouse cancer model is responding to FMD although it displays a weak response to immunotherapy. However, because the focus is on immunotherapy side effects we have now added a lung cancer model, as suggested by the reviewer and confirm the effects on reducing cardiotoxicity (Fig. 8).

6. It will also be interesting to perform their work on patient-derived tumours to further validate their work. The results generated from patient-derived tumours will further support their findings. As reported in a plethora of literatures, the immunity/immune responses from the mouse are different from human, thus testing it in patient-derived tumours will further enhance the knowledge of how diet reduce toxicity.

We agree and would be very interested in collaborating with a clinic that provides patients derived tumors. However, in our experience, only the IRB protocol approval would take at least 6 months, and at this time we have not even identified a hospital ready to provide the tumors.

Minor comments:

1. It seems that the manuscript is finished off in a rush. Some of the phrases require clarity. There are multiple grammatical and typographical errors throughout the manuscript. For example, it stated that “cancer therapeutics with the potential to increase survival.” do the authors mean “increase the overall survival outcomes of cancer patients”? Another example, “preventing the onset of unwelcome side effects.” do the authors mean “unnecessary or unwanted side effects?” Other examples, “Cd45+” in Figure 3; “Figure 6” in Figure 6; “combo therapy” in Figure 1 title; is it anti-PD-L1 or anti-PD-1 in Figure 1 (the manuscript stated combined anti-PD-1 and anti-CTLA4, however, the figure caption stated anti-PD-L1/anti-CTLA-4)?

We have carefully revised the manuscript and corrected the spelling errors found in the paper.

2. The authors exaggerate the toxicity in the heart. It is indeed rare (<5% of cancer patients treated with combination immunotherapies) and does not represent an urgent clinical unmet need.

In fact, cardiotoxicity accounts for 1% of IRAEs cases but 50% of patients with cardiotoxicity die. In addition, the effects of FMD cycles on inflammation and cardiotoxicity, in the presence of increased or neutral effects on tumor growth, is promising since it may reflect a wider effect of FMD against inflammatory side effects each of which may be rare.

We wrote: “Although cardiotoxicity accounts for <1% of IRAEs, the onset of such complications, such as myocarditis, arrhythmia, pericarditis and vasculitis can rapidly degenerate and cause death in 50% of cases”

Overall comment:

If the authors can perform further experiments as suggested above, the manuscript will be suitable for Nature Comm. However, the current manuscript is suitable for Scientific Reports Nature.

Reviewer #2 (Remarks to the Author):

Dr. Cortellino and colleagues perform a study of fasting mimicking diet along with combination immunotherapy regimens in mice with the B16F10 melanoma model. This poorly immunogenic model is made slightly more sensitive to PD-1/OX40 targeted regimens, though not with PD-1/CTLA-4 blockade, with slightly more immune infiltrate (albeit with slightly inconsistent results). This approach also may provide cardioprotection, although again there are some conflicting results here too. I have the following concerns and comments.

1. In the introduction, it is stated that OX40, GITR, 4-1BB targeted treatment improves anti-tumor immunity. It should be noted that while this approach may enhance responses in pre-clinical models, none of these approaches have yet been proven in patients.

We have modified the text following reviewer suggestions.

We wrote: “For example, targeting alternative pathways such as the co-stimulatory molecules OX40, 4-1BB, glucocorticoid-induced TNFR-related protein (GITR) has proven to enhance T-cell mediated

immunity in preclinical models (Croft M., 2003; Pan PY et al., 2002; Watts TH, 2005; Valzasina B et al, 2005; Piconese S et al., 2008), although no clinical studies have confirmed the efficacy of such treatments in humans.”

2. The introduction between lines 66-102 is poorly written and paragraphs are strung together without a coherent narrative. This portion should be rewritten.

We have extensively revised the introduction

3. It seems a bit random to change from combination PD-1/CTLA-4 blockade with 1 FMD cycle to PD-1/OX40 + 2 FMD cycles. Is there a reason for this experimental design?

In general, it is preferable not to administer anti-OX40 and anti-PD-L1 simultaneously as it has been demonstrated by Messenheimer DJ et al. that the simultaneous administration leads to the T lymphocytes hyperactivation and exhaustion, whereas the sequential administration improves the T lymphocytes activation and antitumor efficacy. The anti-PD-1/anti-CTLA4 combination works well together and has no contraindications. The decision to perform 1 cycle of FMD with anti-PD-1/anti-CTLA-4 therapy is due to the fact that the 2 antibodies could be administered together whereas anti-OX40/anti-PD-L1 must be administered sequentially. However we have now also tested 2 cycles of FMD in combination with anti-PD-1/anti-CTLA-4 therapy on lung tumor model, but only observe a trend for improved anti-cancer effects compared to immunotherapy alone

4. Are there effects on the innate immune system with a single cycle of FMD?

One cycle of FMD, with or without anti-PD-1/anti-CTLA-4 treatment, has no effect on innate immune cell population of the melanoma TME. In fact, the immune infiltrate FACS analysis did not reveal significant differences in the M-MDSC (Supplementary Figure 1C), PMN-MDSC (Supplementary Figure 1D), dendritic cells (Supplementary Figure 1E), TAMs population (Supplementary Figure 1F) and M1-M2 TAM polarization (Supplementary Figure 1G, H), both between the various experimental groups and between the standard diet and FMD groups. IHC myeloperoxidase staining of melanoma tumor sections showed no differences in both the number and distribution of myeloid cells within the tumor between the different experimental groups and between the standard and FMD diet groups (Supplementary Figure 2E).

Finally we investigated whether FMD or immunotherapy could affect the B-cell population within the TME. IHC staining of melanoma tumor sections with anti-B220 showed that the B cell infiltrate is very low in the tumor tissue and that it is not affected by FMD or anti-PD-1/anti-CTLA-4 immunotherapy (Supplementary Figure 2D).

5. Figure 4B - it appears that FMD + PD-1/CTLA-4 increases necrosis in the heart compared with PD-1/CTLA-4 blockade alone? The text states the opposite.

We apologize to the reviewer for the error in the graph. We inadvertently inverted the data of the AL PD1/CTLA4 group with those of the FMD PD1/CTLA4 group. We have corrected the error by reporting the correct labeling in the figure (see new Fig. 4B), thanks for reporting the error and sorry again for the error.

For the Reviewer's perusal:

6. Figure 5A-B - it looks like fasting diet increased T cell infiltrations in IgG treated mice. Is this mislabeled?

We agree with the reviewer: FMD slightly increased the staining of CD4 and CD8 lymphocytes in cardiac tissue compared to the IgG group, however, the data are not statistically significant, indicating that treatment with IgG does not significantly affect cardiac lymphocytic infiltrate between AI and FMD groups.

7. The lack of validation in other models is a potential weakness.

In order to establish whether FMD reduces immunotherapy related cardiac adverse events in melanoma or other tumor forms, we tested the effects of FMD and immunotherapy on LLC1 lung tumor, another tumor considered cold because it is unresponsive to immunotherapy. FMD (2 cycles) in combination with anti-OX40/anti-PD-L1 causes a trend for delayed LLC1 lung tumor growth, even if this difference is not statistically significant but only a trend (Figure 8B), while the anti-PD-1/anti-CTLA-4 FMD combination has no effect on tumor growth (Figure 8B). FACS analysis of the immune infiltrate did not reveal significant differences on the percentage of CD4, Tregs and CD8 T lymphocytes, on the CD8 activation (Figure 8C-F). Unlike what has been observed in melanoma, FMD in combination with anti-OX40/anti-PD-L1 increases the percentage of CD8 in the tumor center, as detected by IHC tumor section analysis (Supplementary Figure 4A, B). Notably, Ajona et al (Nature Cancer 2020) had already shown strong effects in enhancing immunotherapy efficacy for more cycles of fasting applied more frequently against this same LLC1 tumor. Because this was already published and we were focusing on cardiotoxicity we did not repeat their treatment method.

The data indicates that, in agreement with the results for the melanoma model, FMD reduces cardiac fibrosis and necrosis (Figure 8G, I, L) by limiting the infiltration of CD3 and CD8 (Figure 8H, M) into the myocardium, and proinflammatory cytokine release (Supplementary Figure 6A, B), thus preventing inflammatory myocardial damage induced by immunotherapy.

Reviewer #3 (Remarks to the Author):

A preclinical study was performed using melanoma-bearing mice treated with two combinatorial ICIs therapies (anti-OX-30 40/PDL-1 or antiCTLA-4/anti-PD-1) during a standard or FMD treatment regimen. Their results indicate that FMD can reduce biomarkers involved in cardiovascular disease without interfering with ICIs therapies. The premise for using FMD in combination with ICI therapy has already been established by the Longo laboratory (Cell Reports 2022). The work presented is of great interest to the immuno oncology field.

General Comments.

1. The method for FMD is not described. Without any knowledge of the methodology, it is difficult to assess its efficacy.

We have added the description of the FMD therapy in the materials and methods.

“The mice underwent 1 or 2 cycles of FMD (4 days each week) starting the third day after tumor implantation and sacrificed on 21 post-injection day. 1 FMD cycle consists of alternating 4 consecutive days of fasting mimicking diet and 3 days of refeeding with standard diet. FMD components are described in Brandhorst et al., (2015), and Di Biase et al., (2016) (Brandhorst et al., 2015; Di Biase et al., 2016). Briefly the day 1 diet provides 1.88 kcal/g (50% of normal daily intake) and is made by a mix of various low-calorie broth powders, a vegetable medley powder, extra virgin olive oil, and essential fatty acids mixed with hydrogel; day 2 diet contains 0.36 kcal/g (10% of normal day intake) and consist of low-calorie broth powders and glycerol mixed with hydrogel.”

2. The study lacks generalizability. The prior study using FMD in combination used a breast cancer model. Now a melanoma model was used. FMD enhanced the anti-tumor efficacy in the breast cancer model but not in the melanoma model. A more comprehensive assessment of the ability of FMD to modulate ICI treatment is needed. Furthermore, making a conclusion in the current study based on one mouse cell line lacks rigor.

We apologize since the way the study was presented gave the impression that FMD cycles do not delay melanoma tumor growth. In fact, FMD plus anti-OX40/anti-PD-L1 was much more effective in delaying melanoma growth compared to anti-OX40/anti-PD-L1 alone although this effects was not significantly different from FMD alone. So clearly, this melanoma mouse cancer model is responding to FMD although it displays a weak response to immunotherapy. We have now tried to make this clearer. To address the reviewers concerns, we have now also added a second cancer model to study the effect of FMD cycles on immunotherapy side effects and cancer growth using (LLC1 lung cancer). As observed for melanoma, FMD (2 cycles) in combination with anti-OX40/anti-PD-L1 causes a non significant trend for delaying the growth of LLC1 lung tumor (Figure 8B), while the anti-PD-1/anti-CTLA-4 FMD combination has no effect on tumor growth (Figure 8B). FACS analysis of the immune infiltrate did not reveal significant differences on the percentage of CD4, Tregs and CD8 T lymphocytes, on the CD8 activation (Figure 8C-F). Unlike what has been observed in melanoma, FMD in combination with anti-OX40/anti-PD-L1 increases the percentage of CD8 in the tumor center, as detected by IHC tumor section analysis (Supplementary Figure 4A, B). Notably, Ajona et al (Nature Cancer, 2020) had already shown strong effects in enhancing immunotherapy efficacy for more cycles of fasting applied more frequently against this same LLC1 tumor. Because this was already published and we were focusing on cardiotoxicity we did not repeat their treatment method.

However, we confirmed that FMD reduces cardiac fibrosis and necrosis (Figure 8G, I, L) by limiting the infiltration of CD3 and CD8 (Figure 8H, M) into the myocardium, reducing proinflammatory cytokine release (Supplementary Figure 6A, B) and preventing the inflammatory myocardial damage induced by immunotherapy.

Specific Comments

1. Page 8, Lines 173-174: results are overstated; the combination of FMD and anti-OX40/anti-PDL1 did not provide a significant benefit against melanoma cell growth.

We made this change based on the concern of the reviewer:

“In this case, two cycles of FMD plus anti-OX40/anti-PD-L1 caused a strong delay in melanoma B16F10 melanoma tumors growth, but most of the effect appears to be caused by the FMD since anti-OX40/anti-PD-L1 did not cause any delay in cancer growth and FMD plus anti-OX40/anti-PD-L1 only caused a non significant trend for improved anti-cancer effects compared to FMD alone”.

2. Page 9, Lines 211—214; results are overstated; there is not a statistical analysis of fibrosis or necrosis

Because we did not report an average value but only the absolute number of mice in which we saw fibrosis or necrosis, we agree with the reviewer that these data should be interpreted with caution and that more detailed studies of AI- and FMD-induced changes in fibrosis and necrosis during ICI therapies should be performed. However, we also quantified validated markers of fibrosis and necrosis including pro-collagen 1 α 1 and metalloproteases in mice myocardial tissue and show that FMD significantly ($p < 0.001$) reduces both pro-collagen and MMP-9 in mice during combinatorial therapies with ICI compared to the AI group, indicating possible antifibrotic and anti-necrotic effects in myocardial tissue. (Figure 4C, 8L)

3. Figure 5: the number of significant digits is not consistently used

- We thank the reviewer for pointing out this error. We have changed the Figure according to this suggestion.

4. Methods section: there is not an analysis to determine whether the studies were adequately powered to detect differences between groups.

The reviewer is correct, we forgot to describe the statistical methods used to predetermine sample sizes. We provide now this information in a paragraph on statistical analysis added in the material and methods section.

“GraphPad Prism 9 was used for statistical analysis and graphing. Differences between three or more groups were analyzed using one-way ANOVA followed by post hoc analysis through Tukey’s multiple comparison test. Comparison of CD3 and CD8 immune cell count in heart sections were performed using the Kruskal–Wallis test with post hoc Mann–Whitney *U*-test. Statistical tests are indicated in the respective sections and figure captions. Differences were considered statistically significant at p value (p) < 0.05 . For in vivo experiments, sample size estimation was performed with G.Power software using a multifactorial variance analysis (ANOVA) repeated measurement, within-between interaction.

A power analysis with significance level $\alpha = 0.05$, (assuming a large effect size $f = 0.4$) indicated that $n=8$ mice per group are needed to achieve a power $(1 - \beta)$ of 0.9 which is considered adequate to detect difference between means at least of two groups.”

.”

REVIEWERS' COMMENTS

Reviewer #1 (Remarks to the Author):

In the revised manuscript, I have no further issues or concerns in the overall revised manuscript. Although I do understand that the time will be the limiting factor to improve the study, I suggest that the authors discuss how single cell expression profiling will further improve the work in the Discussion section, so that researchers who are interested in performing the similar study might consider adding single cell experiment to their work.

Reviewer #2 (Remarks to the Author):

My comments have generally been addressed.

Reviewer #3 (Remarks to the Author):

The authors have adequately addressed the concerns raised.